# A deep learning diagnostic platform for diffuse large B-cell lymphoma with high accuracy across multiple hospitals

Dongguang Li[1], Jacob R. Bledsoe[2], Yu Zeng[3], Wei Liu[4], Yiguo Hu [5], Ke Bi[3], Aibin Liang [6] & Shaoguang Li [1✉]

Diagnostic histopathology is a gold standard for diagnosing hematopoietic malignancies. Pathologic diagnosis requires labor-intensive reading of a large number of tissue slides with high diagnostic accuracy equal or close to 100 percent to guide treatment options, but this requirement is difficult to meet. Although artificial intelligence (AI) helps to reduce the labor of reading pathologic slides, diagnostic accuracy has not reached a clinically usable level. Establishment of an AI model often demands big datasets and an ability to handle large variations in sample preparation and image collection. Here, we establish a highly accurate deep learning platform, consisting of multiple convolutional neural networks, to classify pathologic images by using smaller datasets. We analyze human diffuse large B-cell lymphoma (DLBCL) and non-DLBCL pathologic images from three hospitals separately using AI models, and obtain a diagnostic rate of close to 100 percent (100% for hospital A, 99.71% for hospital B and 100% for hospital C). The technical variability introduced by slide preparation and image collection reduces AI model performance in cross-hospital tests, but the 100% diagnostic accuracy is maintained after its elimination. It is now clinically practical to utilize deep learning models for diagnosis of DLBCL and ultimately other human hematopoietic malignancies.

---

[1] Division of Hematology/Oncology, Department of Medicine, University of Massachusetts Medical School, Worcester, MA, USA. [2] Department of Pathology, University of Massachusetts Memorial Medical Center, Worcester, MA, USA. [3] Department of Pathology, Tongji Hospital of Tongji University School of Medicine, Shanghai, P.R. China. [4] Department of Pathology, The First Affiliated Hospital of Soochow University, Suzhou, Jiangsu, P.R. China. [5] Department of Thyroid Surgery, State Key Laboratory of Biotherpay and Collaborative Innovation Center for Biotherapy, West China Hospital, Sichuan University, Chengdu, Sichuan, P.R. China. [6] Department of Hematology, Tongji Hospital of Tongji University School of Medicine, Shanghai, P.R. China. ✉email: shaoguang.li@umassmed.edu

Precision in diagnosis of human cancers is critical for proposing correct treatment plans that save lives, and histopathology is still a gold standard for diagnosis. Human hematopoietic malignancies are a group of complex diseases and their pathologic diagnoses are generally labor-intensive and challenging. Diffuse large B-cell lymphoma (DLBCL) is a major form of human blood cancer morphologically characterized by a diffuse or sheet-like proliferation of large neoplastic B cells. The diagnosis of DLBCL requires exclusion of other types of lymphomas and hematopoietic tumors that are pathologically similar to DLBCL[1,2]. To accomplish this, the use of immunohistochemistry and/or flow cytometry is routine to increase diagnostic accuracy[1,2]. In particular, expression of B-cell markers such as CD20 and PAX5 in such a diffuse infiltrate of large cells is needed, and exclusion of other B-cell lymphomas with large-cell morphology is mandatory, including mantle cell lymphoma, lymphoblastic lymphoma, and plasmablastic lymphoma, among others[1,2]. It is also necessary to exclude malignant tumors of other histogenesis including carcinoma, melanoma, and sarcoma that may potentially mimic DLBCL[3]. Recently, artificial intelligence (AI) technology has been used in reading pathologic slides of tumor tissues from patients, and the results are promising in reducing labor of pathologists and improving diagnostic accuracy. However, in clinical practice, a high diagnostic accuracy of 100% or >99% is absolutely required to avoid an omission of any patients, but this level of accuracy has not been achieved by any deep learning models so far.

Deep learning is a type of machine learning in which a model learns to perform classification tasks directly from images. Deep learning is usually implemented using neural network architecture[4]. Transfer learning is an approach that applies knowledge of one type of problem to a different but related problem[5]. The use of a pretrained network with transfer learning is typically much faster and easier than the training of a network from scratch, and medical image analysis and computer-assisted intervention problems have been increasingly addressed with deep-learning-based solutions[6]. Although the available deep-learning platforms are flexible, they do not provide specific functionality for medical image analysis and their adaption for this domain of application requires substantial implementation effort[7]. Consequently, there has been substantial duplication of effort and incompatible infrastructure developed across many research groups[8]. On the other hand, acquisition, annotation, and distribution of medical image datasets have higher costs than conduction of many computer vision tasks. For many medical imaging modalities, image generation is costly. Image annotation for many applications requires high levels of expertise from clinicians with limited time. In addition, due to privacy concerns, dataset sharing between institutions is logistically and legally challenging. Although recent tools developed in AI industries are beginning to reduce these barriers, typical datasets remain small. Using smaller datasets increases the importance of data augmentation, regularization, and cross-validation to prevent overfitting. The additional cost of dataset annotation also places a greater emphasis on semi- and unsupervised learning.

Deep learning in computation improves visual object recognition[9] and has exceeded human performance in high-level thinking and reasoning such as playing games[10,11], implying that there would be a great potential for deep learning in improving medicine. In fact, recent studies on pathological analyses of human tissues from patients have shown a promising sign of using deep-learning technology to help pathologists to diagnose human diseases[6,12–17]. In medical practice, an unmet need in computational pathology is to reach high diagnostic accuracy equal or close to 100%.

In this work, we aim to generate highly accurate AI deep-learning models for diagnosing DLBCL. Our results from reading pathologic slides of DLBCL patients from three independent hospitals show that the diagnostic accuracy of our AI models reaches a high level (close to 100%) suitable for clinical use.

## Results

**Establishment of AI models for analysis of pathologic images**. Pathologic tissue slides for DLBCL and non-DLBCL were prepared by taking photographs of the slide images or by scanning the entire slides with a scanner for establishing AI models (Fig. 1a). Non-DLBCL included reactive/non-neoplastic lymph nodes and other types of tumors such as metastatic carcinoma, melanoma, and other lymphomas including small lymphocytic lymphoma/chronic lymphocytic leukemia, mantle cell lymphoma, follicular lymphoma, and classical Hodgkin lymphoma. Unlike other research groups that utilized a single deep CNN, we set up a globally optimized transfer deep-learning platform with multiple pretrained CNNs (GOTDP-MP-CNNs). Specifically, we combined 17 CNNs to utilize them as a whole in generation of our AI models with a goal of increasing the accuracy of the models. For establishment of our AI models, we used 80% of the pathologic images for model training and 10% of the images for model validation. We then used the remaining 10% of the images for model testing (Fig. 1b). Another major feature in generation of our AI models is that we conducted the transfer learning that involved all 17 CNNs to optimize the fitness of the data (Fig. 1b). Therefore, our AI models are expected to be more efficient and accurate than the existing models in disease diagnosis. We need to point out that besides using 0.8–0.1–0.1 (training 0.8, validation 0.1, testing 0.1) during the training, we also tried other ratios such as 0.6–0.2–0.2 and 0.7–0.15–0.15 but did not notice significant differences. This observation indicates that the data split ratio in training does not play a significant role in our deep-learning platform.

**Variations of pathologic images**. During the establishment of our AI models, we faced a difficult challenge in dealing with sample variations. By comparing DLBCL or non-DLBCL images within or across three hospitals, we noticed several major variations reflecting differences in color, morphology, and quality of the slides (Fig. 2a). Furthermore, we reviewed all images and identified more variations, including artificial structures such as air bubbles and empty space (Fig. 2b). In addition, the scanned whole-slide images contained tissues unrelated to DLBCL (Supplementary Fig. 1). We expect that these variations pose an obstacle to establishing highly accurate AI models, which we should intend to overcome.

**Achievement of a high diagnostic accuracy with small datasets**. We aimed to establish AI analytical models with high diagnostic accuracy for DLBCL. Because we anticipated a potential influence of diagnostic accuracy by sample variations across hospitals (Fig. 2) and expected to obtain a limited amount of human samples from each hospital, we thought that it would be realistic to establish single hospital-based AI models with small datasets to determine whether our AI models could reach high diagnostic accuracy. Initially, hematoxylin and eosin (H&E)-stained formalin-fixed paraffin-embedded tissue sections prepared from lymph nodes of DLBCL patients from two unrelated hospitals were correctly labeled by pathology experts and then photographed at ×400 original magnification to produce pathologic images for analysis. Each slide with diagnosis was further confirmed by at least one pathologist with adequate experience. In

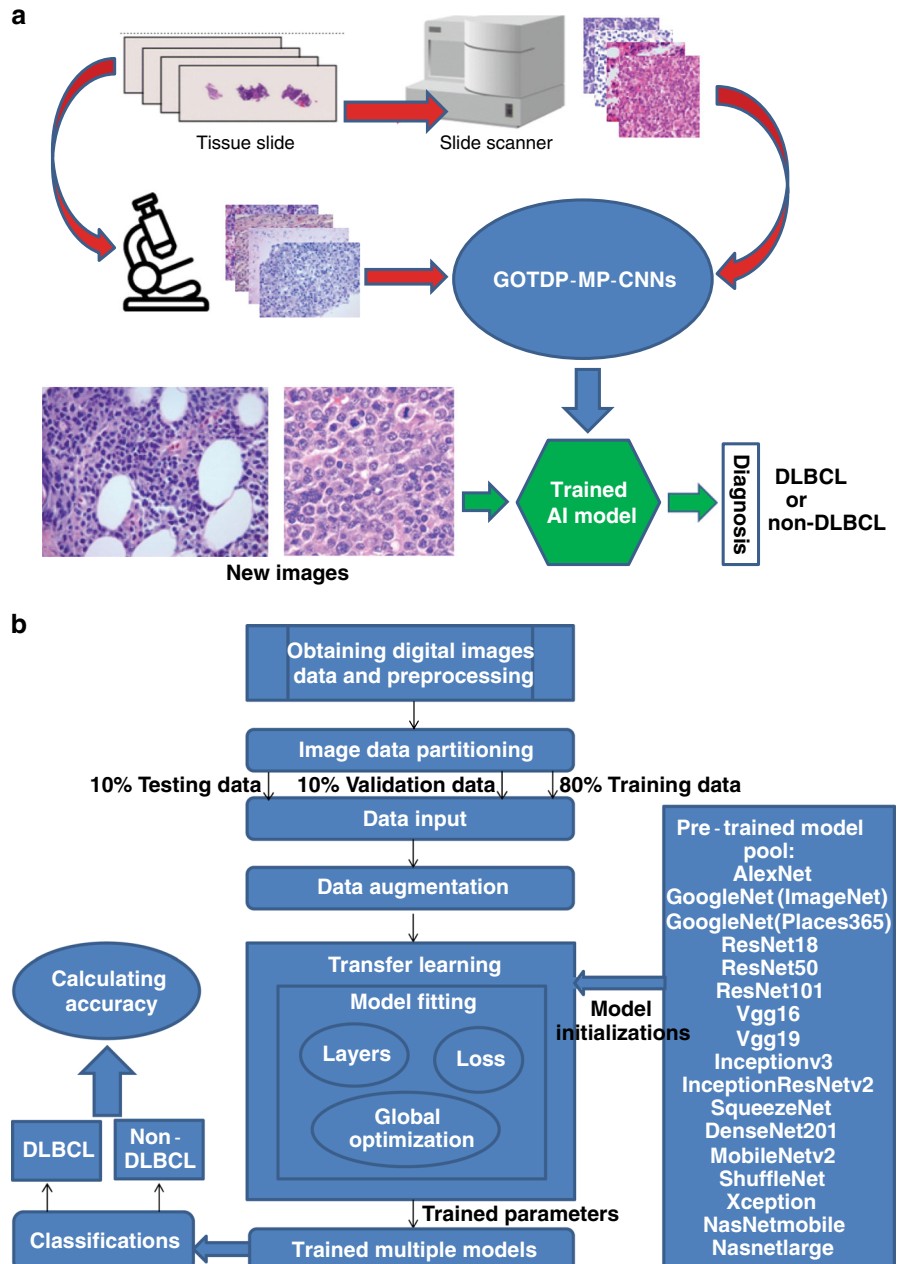

**Fig. 1 Strategies for the establishment of AI models for analysis of pathologic images. a** Workflow of the system. Photographs of the slide images and scanned images of the entire slides for DLBCL and non-DLBCL were used for establishing AI models. **b** Overview of data flow implemented and components of the GOTDP-MP-CNNs. 17 CNNs [AlexNet, GoogleNet(ImageNet), GoogleNet(Places365), ResNet18, ResNet50, ResNet101, Vgg16, Vgg19, Inceptionv3, InceptionResNetv2, SqueezeNet, DenseNet201, MobileNetv2, ShuffleNet, Xception, NasNetmobile, Nasnetlarge] were utilized as a whole in generation of our AI models, and the transfer learning was used to optimize the fitness of the data. A specific AI model was established by training 80% of total samples with 10% of them used for model validation and the remaining 10% of the samples for testing diagnostic accuracy of the established model.

addition, the diagnosis was consistent with the results from other tests such as immunohistochemistry and molecular biology as well as clinical symptoms. Using the samples from each of these two hospitals, hospital-specific AI models were established by training 80% of total samples with 10% of them used for model validation and the remaining 10% of the samples for testing diagnostic accuracy of the established model. From the first hospital (hospital A), 500 DLBCL and 505 non-DLBCL human samples (one pathological image from one patient) were photographed and used. Using combined 17 CNNs (Fig. 1), we tested our AI model for precision by computational reading of randomly mixed pathologic images from hospital A to determine the

percentage of correctly recognizing DLBCL images. Compared with the use of individual CNN, which showed an average percentage of diagnostic accuracy in the three hospitals ranging from 87 to 96% (Table 1), diagnostic accuracy of our AI models reached 100% (Fig. 3a). To confirm the high accuracy of our deep-learning algorithms, we established a AI model for reading pathologic images of DLBCL (number of cases: 204) and non-DLBCL (number of cases: 198) from another hospital (hospital C), where the pathologic images were also collected by photographing (one pathological image from one patient). We found that the diagnostic accuracy of this AI model also reached 100% (Fig. 3a).

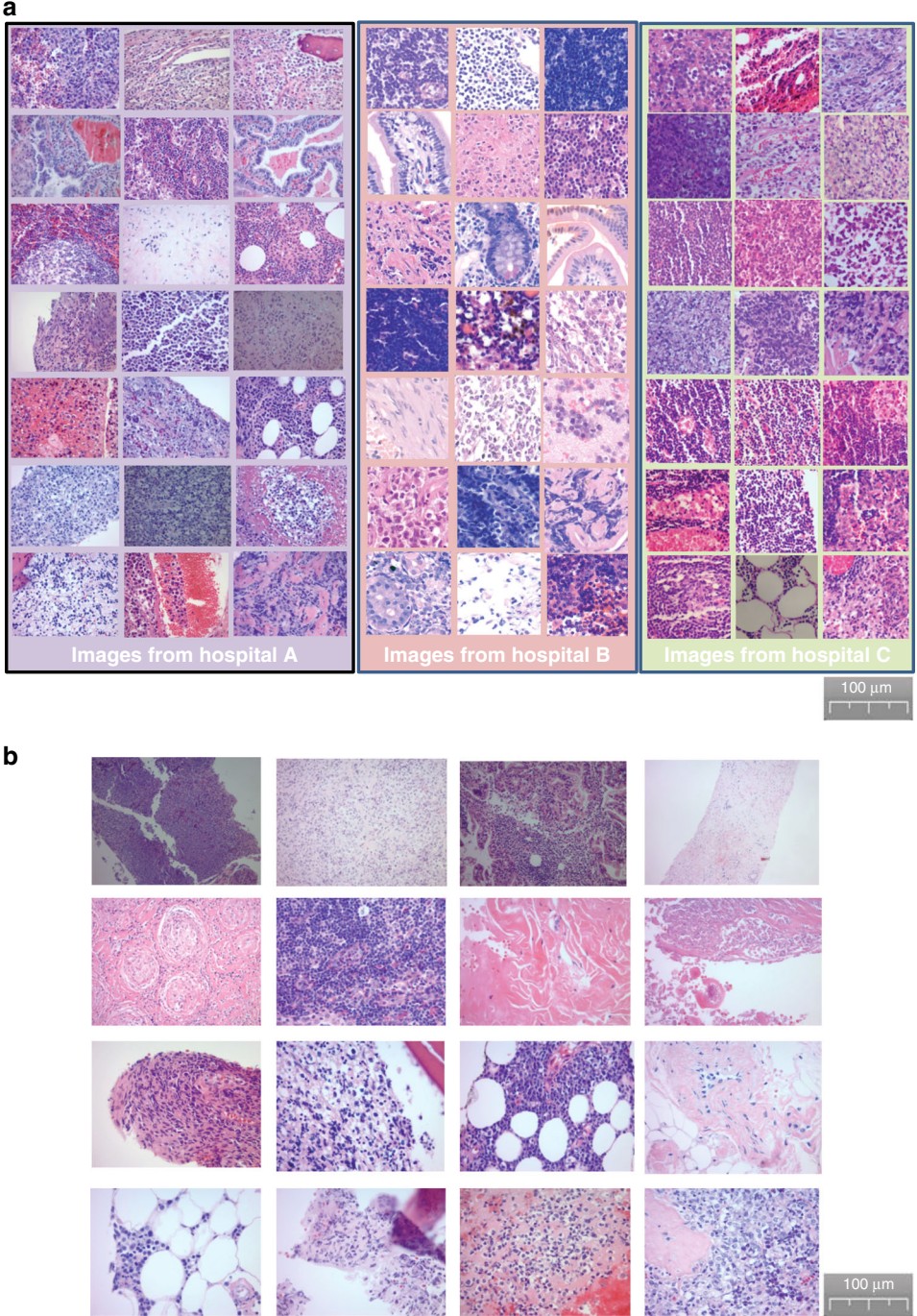

**Fig. 2 Image variations derived from tissue preparation procedures. a** There were some major variations in color, morphology, and quality of the slides, reflecting image variation derived from tissue preparation procedures. **b** There were some artificial structures unrelated to human tissues in the slide images, including air bubbles, empty space, etc.

We next attempted to establish an AI model for reading scanned whole-slide images (WSIs) of DLBCL (1467 square images from 163 cases) and non-DLBCL (1656 square images from 184 cases) samples obtained from the third hospital (hospital B). Nine pathologic images from the tissue areas representative of DLBCL were randomly selected from the scanned image for each patient (Fig. 3b) (nine pathological images from one patient), and a new hospital B-specific AI model was established and tested for its diagnostic accuracy. The diagnostic accuracy reached 99.71%. Unlike hospitals A and C where we used one pathological image from each patient in

building our AI models, we used nine pathological images from each patient for building the model in hospital B. Thus, it is possible that a patient in hospital B has some slide images in the training set and some in the test set. Nevertheless, taking all results from the three hospitals into consideration, we clearly demonstrate that our AI models allow achieving high diagnostic accuracy for DLBCL, building a solid foundation for the use of the AI models in clinical practice for diagnoses of DLBCL and ultimately other hematopoietic malignancies.

On the one hand, it is impossible and unpractical to obtain thousands or more DLBCL patient samples from a single hospital,

**Table 1 Diagnostic accuracy of 17 CNNs for DLBCL.**

| CNNs | Diagnostic accuracy (%) | | |
|---|---|---|---|
| | Model A | Model B | Model C |
| AlexNet | 92.08 | 93.57 | 95.12 |
| GoogleNet | 95.05 | 90.68 | 95.12 |
| Vgg16 | 95.05 | 94.53 | 99.50 |
| ResNet18 | 92.08 | 88.42 | 95.12 |
| SqueezeNet | 92.08 | 89.39 | 92.68 |
| MobileNetv2 | 90.10 | 88.42 | 92.68 |
| Inceptionv3 | 90.10 | 93.89 | 87.80 |
| DenseNet201 | 90.10 | 84.57 | 95.12 |
| Xception | 98.02 | 91.32 | 85.37 |
| Vgg19 | 87.13 | 93.25 | 92.68 |
| Places365GoogleNet | 96.04 | 92.93 | 95.12 |
| InceptionResNetv2 | 94.06 | 96.14 | 96.02 |
| ResNet50 | 86.14 | 90.68 | 87.80 |
| ResNet101 | 89.11 | 91.96 | 97.56 |
| NASNetMobile | 95.05 | 85.21 | 90.24 |
| NASNetLarge | 95.05 | 91.96 | 92.50 |
| ShuffleNet | 87.13 | 88.42 | 85.37 |
| GOTDP-MP-CNNs (with combined 17 CNNs) | 100.00 | 99.71 | 100.00 |

and on the other hand, a tremendous amount of variations can be derived from tissue preparation procedures (Fig. 2). Therefore, we believed that it is critical to be able to use a smaller dataset (<1000 human samples) for establishing an AI model with high diagnostic accuracy. In fact, we were able to establish accurate AI models using limited number of patient samples from each of the three hospitals (Fig. 3a).

The high diagnostic accuracy achieved using our AI models for DLBCL is indeed striking (Fig. 3), which is likely due to the core algorithms we developed. To further validate our AI algorithms, we decided to analyze the dataset CIFAR-10 developed by CIFAR (Canadian Institute for Advanced Research) using our core algorithms, because CIFAR-10 provides a competitive platform for research groups in the world to compare the accuracy of their AI models. Of 49 available independent results, the highest accuracy was 96.53% (Supplementary Table 1). In contrast, the accuracy of our AI model reached 96.88%, explaining why we achieved the high diagnostic accuracy for DLBCL.

**Effect of sample preparation procedures on diagnostic accuracy.** The high diagnostic accuracy of our AI models for reading pathologic slides of DLBCL patients from the three hospitals (hospitals A, B, C) prompted us to test whether a cross-hospital use of the same AI model on DLBCL diagnosis is practical for retaining the high diagnostic accuracy. We were aware of the fact that the sample preparation procedures among the three hospitals differed largely in tissue staining, slide preparation, image collection, etc. We used the AI model established by analyzing the pathologic images of DLBCL patient samples from hospital A, which resulted in a 100% diagnostic accuracy for the patients in that hospital (Fig. 3a), to read those from hospital C. We chose hospitals A and C in this cross-hospital test because in both hospitals, the histomicrographs were taken using microscope cameras. The result showed that the diagnostic accuracy of DLBCL dropped from 100 to 82.09% (Fig. 3c), which is surprising to us because our AI algorithms are among the best (Supplementary Table 1). Then we realized that we did not standardize the shape of the tissue images when conducting this cross-hospital test. The model A was trained with the original rectangular images (the width to height ratio is 4:3) from the hospital A and the tested images from hospital C were squares in shape;

thus, the images from hospital C were twisted and fed into the model A for testing. Therefore, we re-did this test by normalizing the shape of 179 images in hospital C to the shape of the images in hospital A to determine the generalization ability of our AI model. After the shape of the images was unified between the two hospitals, the diagnostic accuracy was significantly increased from 82.09 to 90.50% (Fig. 3c). In other words, the diagnostic accuracy of the model A dropped about 10% (from 100 to 90.50%) in this new cross-hospital test. This 10% drop in diagnostic accuracy suggests that the technical variation introduced by sample preparation procedures between hospitals significantly affects the accuracy. To provide supporting evidence to this assumption, we did two tests. First, we used the model A established in hospital A to read new images (in total 110 images) obtained from the same hospital after the model A was established. Using the images from the same hospital, we would be able to largely eliminate the technical variation introduced by slide preparation methods and imaging equipment. The result showed that diagnostic accuracy for reading the new images remained at 100% (Fig. 3c). Second, we conducted another cross-hospital test, in which we attempted to use our model established in hospital B (model B) to read new tissue images of patients from a different hospital (hospital D) that utilized the slide preparation procedures similar to the ones used by hospital B. For image collection, we scanned the pathologic slides from hospital D to produce whole-slide images (in total 135 images) using the same scanner we used to collect whole-slide images from hospital B. Thus, we basically eliminated the differences between the two hospitals (hospitals B and D) in slide preparation and image collection equipment. We then used the model B established in hospital B to read the images from hospital D, and 100% diagnostic accuracy for DLBCL was achieved in this cross-hospital test (Fig. 3c).

Taken together, these results demonstrate that the technical variability introduced by slide preparation and imaging equipment can be eliminated through standardizing them among different hospitals or building a single hospital-based deep-learning model to achieve a 100% diagnostic accuracy. At present, we believe that a realistic approach is to achieve a high diagnostic accuracy close to 100% by establishing an AI model specifically for each hospital. Because it is not possible to obtain a large number of DLBCL samples from any single hospital, establishment of a highly accurate AI model for DLBCL diagnosis requires using a smaller dataset, which we had achieved (Fig. 3a).

**Sensitivity of AI models.** We intend to have a high standard for using our AI models in clinical practice, and our goal in diagnosis is to reduce false-positive and eliminate false-negative rates. The 100% diagnostic accuracy in hospital A (Fig. 3a) and hospital C (Fig. 3a) indicated that both false-positive and false-negative rates for diagnosis of DLBCL were zero. However, the diagnostic accuracy for hospital B was 99.71% (Fig. 3a), suggesting a possibility that a false-negative case existed, although the false-positive rate was zero. In clinical diagnosis using AI models as an initial screening tool for pathologists to skip reading of a significant number of non-DLBCL slides, false-positive cases would be acceptable but false-negative cases are absolutely not. Therefore, we carefully examined that one case which was diagnosed as DLBCL by pathologists but was not recognized as DLBCL by our AI model (Fig. 4a). We thoroughly analyzed clinical data of this patient and found that the patient had a poor response to conventional therapy for DLBCL and did not have typical clinical symptoms for DLBCL. To us, it was questionable whether this patient really had DLBCL. Based on the symptoms and therapy response, we came up with a conclusion that this patient was unlikely to have typical DLBCL. Therefore, we traced back the

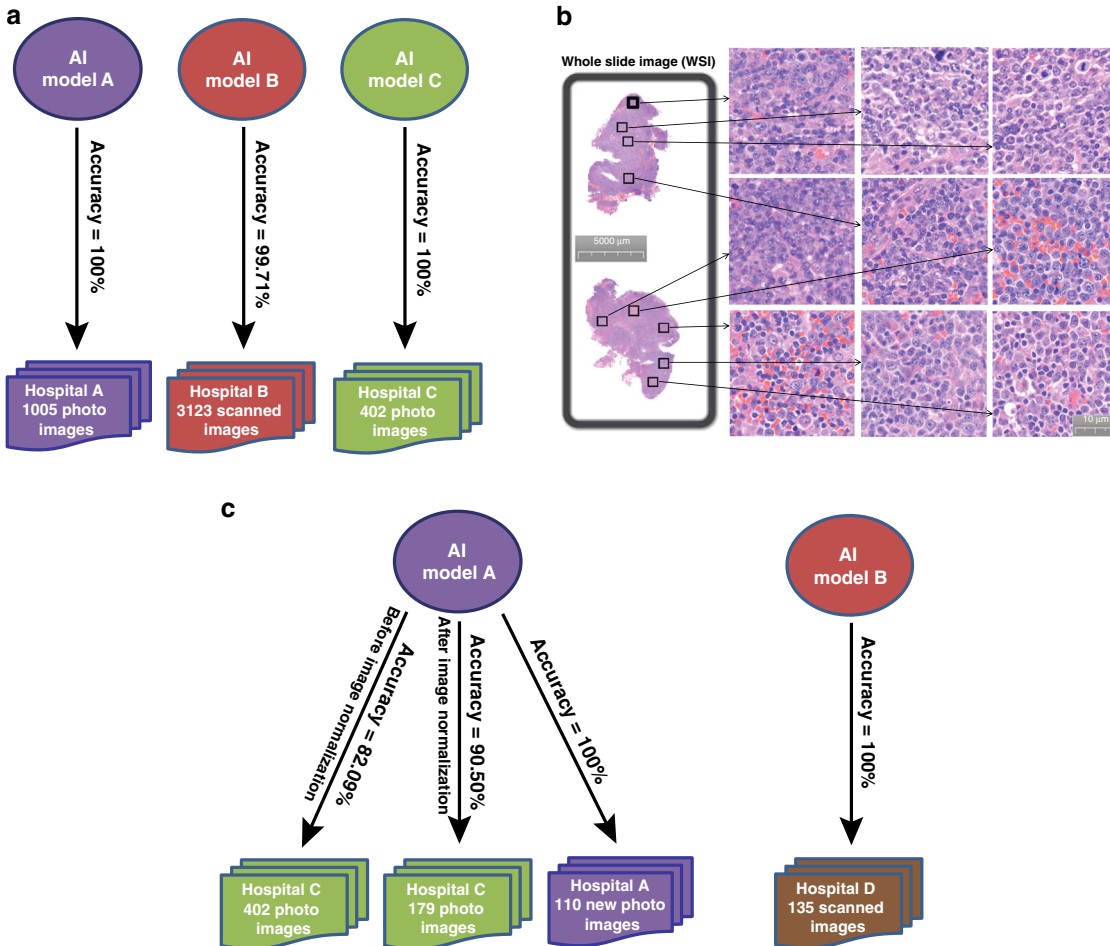

**Fig. 3 A high diagnostic accuracy of our AI models with the use of smaller datasets. a** Hematoxylin and eosin (H&E)-stained formalin-fixed paraffin-embedded tissue sections prepared from lymph nodes of DLBCL and non-DLBCL patients from four unrelated hospitals (hospitals A, B, C) were photographed (hospitals A, C) or scanned (hospital B) at ×400 original magnification to produce pathologic images for generating three separate AI models (Models A, B, C), each of which was specifically generated using the DLBCL and non-DLBCL samples from the corresponding hospital. A high diagnostic accuracy was reached by the three AI models (100% for hospital A, 99.71% for hospital B, 100% for hospital C, respectively). **b** Analysis of whole-slide images from hospital B from each patient by randomly selecting nine pathologic images within the DLBCL cell-containing areas. Thus, each experiment was done nine times. **c** The diagnostic accuracy dropped from 100 to 90.50% or 82.09% with or without unifying the shape of the images between hospital A and hospital C when cross-hospital use of the deep-learning model A was carried out to read the slide images of patients from hospital C. The diagnostic accuracy increased to 100% when the model A was used to read new images of patients in the same hospital. 100% diagnostic accuracy was also achieved when the model B was used to reach the slide images of patients from a new hospital (hospital D) after elimination of the technical variability introduced by slide preparation procedures and image collection equipment.

diagnostic history of this patient and found that the original diagnosis was follicular lymphoma or follicular diffuse mixed type, which could progress to become DLBCL at a later stage of the disease. Until recently, this patient developed DLBCL-like disease, and the pathologic images we analyzed actually reflected the tissues collected from the patient prior to transitioning to DLBCL. If we eliminated this case in our analysis, which we should have, the diagnostic accuracy for DLBCL in hospital B would have reached 100%. Because of this incidence, we decided to collect additional DLBCL and non-DLBCL lymph node samples from hospital B (in total 531 400× images) to further test the diagnostic accuracy of our AI model established previously in this hospital. To be careful, we compared the diagnostic accuracy of our AI model with that of seven invited pathologists for reading the same set of DLBCL and non-DLBCL 400× images. The pathologists used about 60 min on average to read all 531 images, and the highest diagnostic accuracy among the seven pathologists was 74.39% (Fig. 4b). In contrast, it took <1 min with a notebook computer for our AI model to finish reading all these images, and

the diagnostic accuracy was 100%. We should point out that pathologists often read tissue slides at the magnitude of 400× and lower with help from other clinical tests such as immunohistochemistry and molecular biology to increase the accuracy of pathological diagnosis. Thus, we do not expect a 100% diagnostic accuracy from the pathologists when they only read H&E-stained pathologic tissue slides at one magnification. Regardless, these results demonstrate that our AI models for DLBCL diagnosis have exceeded performance by pathologists and reached 100% diagnostic accuracy and sensitivity for DLBCL with almost no false-positive and false-negative errors.

## Discussion
This work presents the GOTDP-MP-CNNs platform for deep learning in medical imaging. Our modular implementation of the typical medical imaging machine learning pipeline allows us to focus our implementation effort on specific innovations, while leveraging the work of others for the remaining pipeline. The GOTDP-MP-CNNs platform provides implementations

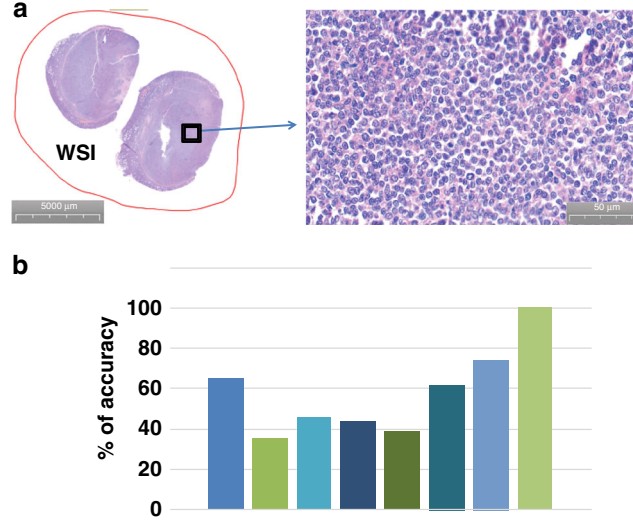

**Fig. 4 Sensitivity of AI models. a** Pathologic images of the sole case that was diagnosed as DLBCL by pathologists but was not recognized as DLBCL by our AI model. **b** Comparison of diagnostic accuracy of DLBCL between pathologists and our AI model by reading additional DLBCL and non-DLBCL lymph node samples (in total 531 400× images) from hospital B. Seven experienced pathologists were invited to read the images. The pathologists used about 60 min on average to read all 531 images with the highest diagnostic accuracy of 74.39%. In contrast, the diagnostic accuracy of our AI model was 100%.

for data loading, data augmentation, network architectures, loss functions, and evaluation metrics that are tailored for the characteristics of medical image analysis and computer-assisted intervention.

Transfer learning is an efficient solution for many problems. The classification accuracy on the ImageNet validation set is the most common way to measure the accuracy of networks trained on ImageNet. Networks that are accurate on ImageNet are also often accurate when one applies them to other natural image data sets using transfer learning or feature extraction. However, high accuracy on ImageNet does not always transfer directly to other tasks, so we believe that it is a better idea to use an ensemble of multiple networks as we used in our study.

Limited AI study on DLBCL diagnosis is available, and one report shows a 95% diagnostic accuracy for DLBCL in an intra-hospital test[18]. In clinical practice, a diagnostic accuracy of 100% or >99% is critical and this level of accuracy has not been reported for pathological diagnosis of any types of hematopoietic malignancies based on only reading H&E-stained pathologic tissue slides. In this study, we have established AI deep-learning models for diagnosing DLBCL by only reading the H&E-stained pathologic slides at ×400 magnification with a 100% accuracy in multiple hospitals. Our current success in AI-assisted reading of pathologic slides of DLBCL with the high diagnostic accuracy paves a road to beginning to employ AI deep-learning models in diagnostic histopathology of human hematopoietic malignancies. Because the technical variation introduced by tissue slide preparation and image collection causes a significant reduction in the diagnostic accuracy of AI models on cross-hospital use, one practical strategy to overcome this problem is to standardize slide preparation procedures and image collection equipment among all hospitals, although this approach is challenging. Another practical strategy is to establish "customized" AI models for a particular hospital where the same slide preparation procedures

and image collection equipment could be employed, but this requires an ability to establish highly accurate AI models using smaller datasets obtainable from a single hospital. Although this is a difficult approach, its feasibility has been shown by establishing our AI models with 100% diagnostic accuracy for DLBCL using <1000 human samples. On the other hand, 100% sensitivity on DLBCL diagnosis indicates that our AI models reduce false-positive and false-negative errors to a level close to zero. We believe that it is time to employ AI models for pathologic diagnosis of DLBCL to reduce workload of pathologists and soon for DLBCL subtype classification and other hematopoietic malignancies.

## Methods

**Ethical approval**. All tissue slides involved in this study were historical samples that were photographed or scanned and de-identified before inclusion in the study. Therefore, approval by the Institutional Review Board is not required.

**Slide preparation and validation**. Throughout our study, only lymph nodes were used for generating tissue slides and images. Lymph node excision and biopsy samples from DLBCL and non-DLBCL specimens were prepared using routine processing procedures including formalin fixation, automated processing, and paraffin embedding, followed by sectioning at 5 microns and H&E staining using automated strainers. DLBCL diagnosis was established based on the current WHO classification including H&E morphology supplemented by immunohistochemistry and/or flow cytometry[1]. Diagnosis of all cases was performed by experienced board-certified hematopathologists. Non-DLBCL diagnoses included various causes of benign reactive lymphadenopathy, metastatic tumors to lymph nodes including carcinomas and melanomas, and other lymphomas including T-cell lymphomas and other B-cell lymphomas including small lymphocytic lymphoma/chronic lymphocytic leukemia, mantle cell lymphoma, and follicular lymphoma, among others. H&E slides were photographed at ×400 original magnification using microscope-based cameras and saved in jpeg format without modification of the photomicrographs in any way. In detail, the slide images were collected from three hospitals in the following ways:

Photos taken from hospital A:
- Objective magnification: ×40
- Original image: 2592 × 1944 pixels
- Original image file size: 14.4 Mb
- Pixel size: 2.2 μm × 2.2 μm

Whole slides information from hospital B:
- Average slide dimensions in pixels: ~200,000 × ~400,000
- Average file size: ~10 Gb
- Objective magnification: ×40
- Pixel size: 0.121547 μm × 0.121547 μm
- Cropped image for classification at 945 × 945 pixels

Photos taken from hospital C:
- Objective magnification: ×40
- Original image: 2048 × 1536 pixels
- Original image file size: 5–8 Mb
- Pixel size: 3.45 μm × 3.45 μm
- Cropped image at 1075 × 1075 pixels

**Core algorithms**. In order to achieve high diagnostic accuracy, we developed a Globally Optimized Transfer Deep-Learning Platform with Multiple Pretrained CNNs (GOTDP-MP-CNNs) that provides a modular deep-learning pipeline for a range of medical imaging applications. Components of the GOTDP-MP-CNNs pipeline, including data loading, data augmentation, network architectures, loss functions, and evaluation metrics, are showed in Fig. 1b. GOTDP-MP-CNNs were built on the MATLAB-R2019a framework, supporting features such as visualization and computational graphs by default.

For the AI classification of DLBCL, we used deep-learning approaches implemented with CNNs (or ConvNet), one of the most popular algorithms for deep learning with images. In considering applying deep-learning methods to medical images, we took advantage of the following:

(1) A CNN can be trained to independently predict a disease (such as DLBCL in this study) with a reasonable high accuracy, and a trained CNN can identify predictive imaging features for a given pathologic slide image;

(2) a pretrained CNN built by experts can be retrained to perform new recognition tasks using a technique called transfer learning;

(3) while many pretrained CNNs are trained on more than a million high-resolution images to recognize 1000 different objects, accurate transfer learning can be achieved with much smaller datasets;

(4) by applying a true global optimization algorithm in selecting the best training options of transfer learning, a retrained CNN can perform better than other algorithms;

(5)  jointly using multiple different fine retrained models in classifying medical images, the developed GOTDP-MP-CNNs trained with small datasets can achieve near 100 % of diagnostic accuracy upon reading pathological images for a human disease (such as DLBCL in this study); and

(6)  the developed GOTDP-MP-CNNs can be widely used in medical images classification for other diseases.

**Global optimization algorithms**. Usually to conduct transfer learning we should test the performance of the newly trained network. If it is not adequate, typically we should try adjusting some of the training options and retraining. Obviously, there is no guarantee to achieve the best performance in this way. To overcome the problems and achieve the highest accuracy in classification, in retraining, we not only use multiple models but also adopt our true global optimization algorithms (SDL)[19]. The SDL global optimization algorithms were developed in DNA microarray data analysis[19], which is a collection of strategies in searching for a global minima or maxima of a multiple variables equations. Minimization of the loss was achieved via SDL, and the final loss was the weighted average of the losses over a model pool.

**Pretrained convolutional neural networks**. Pretrained Neural Network Models used in this study include Alexnet, Googlenet(ImageNet), Goolgenet(Places365), Resnet18, Resnet50, Resnet101, Vgg16, Vgg19, Inceptionv3, Inceptionresnetv2, Squeezenet, Densenet201, Mobilenetv2, Shufflenet, Xception, Nasnetmobile, and Nasnetlarge (Fig. 1b). The network overview and details are provided in the Online Appendix.

Pretrained networks have different characteristics that matter when choosing a network to apply to a given problem. The most important characteristics are network accuracy, speed, and size. Choosing a network is generally a tradeoff between these characteristics.

To improve classification results, usually one tries those available pretrained networks built by experts one by one with the hope of achieving higher accuracy by luck. The developed GOTDP-MP-CNNs do not compare and/or choose pretrained networks beforehand. Instead, it takes advantage of every pretrained model to achieve much better performance than any single model.

Software infrastructure for general-purpose deep learning is an additional development. Due to the high computational demands of training deep-learning models and the complexity of efficiently using modern hardware resources (general-purpose graphics processing units and distributed computing, in particular), numerous deep-learning libraries have been developed and widely adopted.

**General system**. Deep convolutional neural networks (CNNs) have emerged as an important image analysis tool. The ability of CNNs to learn predictive features from raw image data is a paradigm shift that presents exciting opportunities in medical imaging. In order to achieve very high diagnostic accuracy, we developed a Globally Optimized Transfer Deep-Learning Platform with Multiple Pretrained CNNs (GOTDP-MP-CNNs) that provides a modular deep-learning pipeline for a range of medical imaging applications. Components of the GOTDP-MP-CNNs pipeline including data loading, data augmentation, network architectures, loss functions, and evaluation metrics are shown in Fig. 1b.

**Hardware and software**. All experiments were conducted on a CPU Sever (Intel® Core™ i5-8250U CPU@1.80 GHz Installed Ram 8.00GB) and a Laptop computer (Microsoft Surface Pro: Intel® Core™ i7-4650U CPU @1.70 GHz 2.30 GHz RAM8.00GB Windows 8.1). In particular, the MATLAB2019a was used for training AI models. We took advantage of a few toolboxes provided by MATLAB, such as the deep-learning toolbox and the image processing toolbox, in data preparation, programming, and deployment. GOTDP-MP-CNNs were built on the MATLAB-R2019a framework, supporting features such as visualization and computational graphs by default.

**Model testing statistics**. We use the following definitions:
DLBCL: positive for DLBCL
Non-DLBCL: negative for DLBCL (healthy or other diseases)
True positive (TP): the number of cases correctly identified as DLBCL
False positive (FP): the number of cases incorrectly identified as DLBCL
True negative (TN): the number of cases correctly identified as healthy or other diseases
False negative (FN): the number of cases incorrectly identified as healthy or other diseases
Accuracy: The accuracy of a test is its ability to differentiate the DLBCL and Non-DLBCL cases correctly. To estimate the accuracy of a test, we should calculate the proportion of true positive and true negative in all evaluated cases. Mathematically, this can be stated as:

$$\text{Accuracy} = (\text{TP} + \text{TN})/(\text{TP} + \text{TN} + \text{FP} + \text{FN}) \quad (1)$$

Sensitivity: The sensitivity of a test is its ability to determine the DLBCL cases correctly. To estimate it, we should calculate the proportion of true positive in

DLBCL cases. Mathematically, this can be stated as:

$$\text{Sensitivity} = \text{TP}/(\text{TP} + \text{FN}) \quad (2)$$

Specificity: The specificity of a test is its ability to determine the Non-DLBCL cases correctly. To estimate it, we should calculate the proportion of true negative in Non-DLBCL cases. Mathematically, this can be stated as:

$$\text{Specificity} = \text{TN}/(\text{TN} + \text{FP}) \quad (3)$$

Prediction scores: To classify an input image into one of two classes (DLBCL and Non-DLBCL), a neural network has an output layer of 2 neurons, one for each class. Passing the input through the network results in calculating a numeric value for each of those neurons. These numeric values, called as prediction scores, represent the network's prediction of the probability of the input belonging to each class.

**Dataset curation**. The datasets were not curated, because we intended to test the applicability of the proposed system in a real-world, clinical scenario. Across all datasets, no slides were removed from the collections from the hospitals. Specifically, although we identified whole-slide images with poor image quality arising from imaging artifacts or tissue processing (Fig. 2), we did not remove any images containing tissue-processing artifacts from analysis.

**Transfer learning**. We took 17 pretrained image classification networks that have already learned to extract powerful and informative features from natural images and used them as a starting point to learn our new task. The majority of the pretrained networks were trained on a subset of the ImageNet database[20]. These networks have been trained on more than a million images and can classify images into 1000 object categories. Fine-tuning pre-networks with transfer learning is often faster and easier than constructing and training new networks.

**Training options**. The learning rate varied from 0.01 to 0.0001. We used mini-batches of size 32 to 128 for 17 pretrained models. We specified the maximum number of epochs as 30 and validation frequency as 20, respectively. All selected pretrained models were initialized with ImageNet pretrained weights and biases. Three optimization algorithms (SGDM, RMSProp, and Adam) were used to minimize the loss functions.

**Slide diagnosis**. We retrained 17 pretrained models, and no best-performing model on the validation set was selected. By contrast, a last classification layer computed the weighted scores, a score matrix $S$, for both DLBCL and Non-DLBCL. The score matrix $S$ consists of two vector elements, $S_{\text{DLBCL}}$ and $S_{\text{Non-DLBCL}}$.

$$S = \begin{bmatrix} S_{\text{DLBCL}} \\ S_{\text{Non-DLBCL}} \end{bmatrix} = \begin{bmatrix} S_{1,1} & S_{1,2}...S_{1,N} \\ S_{2,1} & S_{2,2}...S_{2,N} \end{bmatrix} \quad (4)$$

Where $N$ is the number of models, and $S(i,j)$ are the vector elements of scores for each class (in this study, $i = 1, 2$ and $j = 1, 2, …, 17$).
We also define $\text{TS}_{\text{DLBCL}}$ and $\text{TS}_{\text{Non-DLBCL}}$ as follows:

$$\text{TS}_{\text{DLBCL}} = 1/N \sum_{n=1}^{N} S_{\text{DLBCL}}(n) \quad (5)$$

$$\text{TS}_{\text{Non-DLBCL}} = 1/N \sum_{n=1}^{N} S_{\text{Non-DLBCL}}(n) \quad (6)$$

Here $\text{TS}_{\text{DLBCL}}$ is the final score of DLBCL and $\text{TS}_{\text{Non-DLBCL}}$ is the final score of Non-DLBCL.

$$\text{Diagnosis} = \begin{cases} \text{DLBCL}, & \text{if } \text{TS}_{\text{DLBCL}} \geq \text{TS}_{\text{Non-DLBCL}} \\ \text{Non-DLBCL}, & \text{if } \text{TS}_{\text{DLBCL}} < \text{TS}_{\text{Non-DLBCL}} \end{cases} \quad (7)$$

**CIFAR-10 dataset**. The classification accuracy on the ImageNet validation set is the most common way to measure the accuracy of networks trained on ImageNet. The CIFAR-10 based on the ImageNet, developed by CIFAR (Canadian Institute for Advanced Research), are datasets of RGB images with its classification labeled commonly used in object recognition. It is widely used for image classification task/ benchmark in research community.

The CIFAR-10 dataset contains 60,000 32 × 32 color images in 10 different classes, with 6000 images per class. There are 50,000 training images and 10,000 test images. The publicly shared CIFAR-10 dataset is available at http://rodrigob. github.io/.

The 10 different classes represent airplanes, cars, birds, cats, deer, dogs, frogs, horses, ships, and trucks. For the time being the published highest classification accuracy achieved is 96.53% (http://rodrigob.github.io/are_we_there_yet/build/ classification_datasets_results.html). We explored the CIFAR-10 with our GOTDP-MP-CNNs and 96.88% accuracy is reached.

**Reporting summary**. Further information on research design is available in the Nature Research Reporting Summary linked to this article.

## Data availability

We present three illustrative medical image analysis applications built using GOTDP-MP-CNNs infrastructure. All of the raw image data can be downloaded from www.umass.edu/AI/data upon request to shaoguang.li@umassmed.edu. This paper was produced using no publicly available DLBCL imaging data except the experimental CIFAR-10 data. The authors have made every effort to make available links to these resources as well as make publicly available the software methods and information used to produce the datasets, analyses, and summary information. Further information on research design is available in the Nature Research Reporting Summary linked to this article. All data supporting the findings of this study are available within the paper or from https://fts.umassmed.edu (user name: dli; password: Dong1956) or from the corresponding author upon reasonable request to shaoguang.li@umassmed.edu. The size of our research data is too huge to be properly accepted and stored in public repositories. Also, due to the complexity of our research data, it is better for public users to reach out to shaoguang.li@umassmed.edu for avoiding any misunderstanding of the data and for using the data appropriately.

## Code availability

Whole source code can be found from https://fts.umassmed.edu (user name: dli; password: Dong1956) or obtained by sending a request to shaoguang.li@umassmed.edu.

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

## Acknowledgements

This work was supported by grants from the National Institutes of Health (R01CA176179, R01CA222590, R21CA209298) to S.L. Y.Z. was supported by a Clinical Research Cultivation Project of Shanghai Tongji Hospital [ITJ (QN) 1907].

## Author contributions

D.L. and S.L. conceived the study. D.L., J.R.B., Y.Z., W.L., K.B., and S.L. prepared patient samples and pathologic images. Y.H. and A.L. helped to obtain patient samples. J.R.B. helped to prepare the manuscript. D.L. and S.L. wrote the manuscript.

## Competing interests

The authors declare no competing interests.
