## [Peer Review File · Nature Communications]

Reviewers' Comments:

Reviewer #1:

Remarks to the Author:

Li et al are presenting a deep learning platform including multiple convolutional neural networks for the diagnosis of human diffuse large B cell lymphoma (DLBCL) from histopathology slides of the lymph. Using data from three different hospitals, their method achieved diagnostic rate of 100% for hospital A, 99.71% for hospital B and 100% for hospital C respectively. However, none of the three models is generalizable, which significantly limits the impact and novelty of this work. Previous studies have shown that deep learning models for histologic diagnosis can be generalized in several cancer types.

Major Comments

1. Combining multiple classifiers is not a novel idea, and also the authors need to compare the performance of their approach to each individual CNN. Also, how robust are the results to the data split (training, validation, testing)?
2. The authors can build a model using the data from Hospitals A and C together since the images are obtained with the same method. This could address the sample variation since now the model has taken the variations into account during training. This could potentially yield a more generalizable model.
3. Some of the data, as stated in line 98, were obtained by taking photos of slides and some by scanning them. Is there any difference at the resolution? At which resolution were the whole slides from Hospital B scanned?
4. The order of presenting the results is a little confusing. It is not clear that the authors built 3 different models (one for each hospital) because of the difference in the sample preparation and collection (microscope camera vs scanned) until the paragraph "Effect of sample preparation procedures on diagnostic accuracy". It will be better to re-write/merge this paragraph with the previous one "Achievement of a high diagnostic accuracy for clinical use" and maybe the next one "Establishment of AI models using small datasets" to improve the flow of the paper.
5. Regarding the whole slide model, it seems that the authors used parts of the image to build their model (line 176). Since the CNNs mentioned in Fig 1 have the capacity to tile the images why wasn't the whole image used?
6. To better support the claim in line 183 about the "unprecedented" high diagnostic accuracy of their models, it is important that the authors address fully in their introduction the performance of the current and other published methods to diagnose DLBCL.
7. The CIFAR-10 dataset is a collection of slides with labels such as "airplanes, cars etc". The GOTDP-MP-CNN platform generated by the authors outperformed all other architectures included in the CIFAR-10 platform in classifying these images. I am not sure if this comparison is relevant to DLBCL diagnosis.
8. The comparison with the pathologists' performance is interesting and indeed it is demonstrating the network's high performance. Is looking at 400x images only the common practice for pathologists to make a diagnosis? Do they also look at IHC results? I want to make sure that the network's performance is compared to the actual state-of-the-art practice in the clinic. Also, are the 7 pathologists experienced in DLBCL diagnosis?

9. In line 277 is the first time we see the term "GOTDP-MP-CNN" used in the paper although it appears in Fig1 without explanation except for the legend. It would be helpful if the term appeared earlier in the manuscript.

10. Will the images be made available to the scientific community?

Minor Comments

1. Regarding the title of the study, it might be a little redundant to have both terms "artificial intelligence" and "deep learning".

2. In line 75, the word "Acquisition" should probably be lowercase.

3. In Fig1b, would be more aesthetically pleasing to align all the squares properly.

4. In Fig3 I don't see a legend for panel b.

5. Line 188: probably meant to say "In contrast, "

6. Fig 4a. The little yellow box is unreadable and may need to be removed or enlarged.

Reviewer #2:

Remarks to the Author:

I am not expert in AI systema as I am an haemopathologist, therefore my objections are based only on the histopathology side of the story. I accept the AI can be great, I have no way to judge that but I am worried as far as the handling of the pathology side is concerned.

I have several worries.

Against what the authors measure the accuracy? How were diagnosed the cases used as DLBCL or not-DLBCL to train the system? Which immunohistochemical panels were performed? Which was the anatomical location of the samples: only lymph nodes or also other organs?

DLBCL is an heterogeneous group of lymphomas. What about the T cell rich B cell lymphomas, the Burkitt's lymphoma , the high grade B cell lymphoma, the double hit DLBCL and the post transplant DLBCL, just to mention the main types.

Was the non-DLBCL group composed by the same diagnosis in all the hospital?

The authors do not provide a list of these diagnosis of the cases used as non-DLBCL. This is a critical omission as the composition of this list is critical to properly train the system.

Variation in quality of histological preparation among hospital is a very well-known problem.whether. Ther authors could have started to study it, for example by repeating the analysis after cutting and staining all the case just in one hospital: this simple measure could already improve diagnosis across hospitals. Another cause could have been the different composition of the non DLBCL list in different hospitals.

This variability makes this system of scarce practical utility as hemopathology is usually performed in centralised referral centres that cover many different hospitals.

To investigate the feasibility of obtaining an AI system in each hospital the authors worked on a set of 24 DLBCL patients and 35 non-DLBCL individuals. I am an hemopathologist, not a statistician or a computer scientist, however a set of 35 patients other than DLBCL does not even cover the number of entities other than DLBCL normally seen by an haemopathologist.

When the author discuss the detection of a false negative diagnosis and review the case, it is unclear

why they decided that the case was not a DLBCL. They do not provide enough information, in particular immune profile, proliferating fraction and proliferating fraction distribution in order to properly conclude that the case was not a DLBCL. The diagnosis of "follicular diffuse mixed type" (probably what Lennert called "diffuse centrocytic centroblastic") is now mostly recognised to be DLBCL. Alternatively it can represent the extrafollicular component of a follicular lymphoma but for a pathologist to make a mistake is pretty bad!

The authors write:

"To be careful, we compared the diagnostic accuracy of our AI model with that of seven invited pathologists for reading the same set of DLBCL and non-DLBCL 400x images. The pathologists used about 60 minutes in average to read all 531 images, and the highest diagnostic accuracy among the seven pathologists was 74.39% (Fig. 4b). In contrast, it took less than 1 minute with a notebook computer for our AI model to finish reading all these images, and the diagnostic accuracy was 100%."

I guess they mean that the pathologists only used an HE. But I hope that the AI system was trained using diagnosis done with the support of immunohistochemistry. This experiment to me make no much sense. The gold standard is to make the DLBCL diagnosis with immunohistochemistry, not only to verify the B nature of the malignancy but also to establish subtypes (like e.g. Germinal Centre vs non-Germinal Centre type, Burkitts; expression of cMyc).

Furthermore to give 60 minutes to look at 531 cases is non sensical: of course at that speed an human pathologist will fail, is a self-fulfilling prophecy. Plus it is misleading to test the system against diagnosis done only on HE: this is not the normal clinical practice.

The authors write "In clinical practice, a diagnostic accuracy of 100% or greater than 99% is critical and this level of accuracy has not been reported for pathological diagnosis of any types of hematopoietic malignancies."

This is rather inaccurate and not supported by any evidence. For example I had a very quick look at our double reporting records and, to a very rough analysis, we have an agreement between two or more pathologists on the diagnosis of DLBCL of 98.35, of

98.7% in the diagnosis of Follicular lymphoma

Minor issues include:

Line 150: why use only 400x? It could miss the presence of some low grade component in the same slides

Reviewer #3:

Remarks to the Author:

In their manuscript, Li et al. propose a deep-learning-based platform trained for the diagnosis of diffuse large B cell lymphoma (DLBCL) and non-DLBCL from pathologic images.

With the technique proposed, the authors claim to achieve intra-dataset accuracies between 99.7-100% on data from 3 different hospitals, and inter-dataset accuracies between 80-82%.

The novelty resides in using a combination of 17 trained convolutional neural networks (CNN), instead of comparing performance and choosing the best one as often done.

They claim it's the first time a perfect accuracy is reached, which, the way it is stated, is incorrect (see for example, "Deep Convolutional Neural Networks Enable Discrimination of Heterogeneous Digital Pathology Images"). Better setting of the state of the art could be done and the claim phrased more

carefully and more precisely; 100% is achieved on intra-dataset test set.

While the results are impressive and the method very interesting, the method lacks details to allow readers be convinced. More specifically, the code should be made available to the reviewers (and later to the readers) so that we could properly assess the work. I can't comment on the well documented sources.

A few important details needed that would help the readers would be:

- * Transfer learning is used: what were the networks initially trained on? Where they all pre-trained one after the others with their own outputs, or the 17 networks trained as one with a "merged" output?
- * The authors mention variations of pathologic images between hospitals and also presence of artifacts, and they: "We expect that these variations pose an obstacle to establishing highly accurate AI models, which we should pay much attention to when generating our AI models." but don't explain properly how they dealt with it
- * line 150 "were correctly labeled by pathology experts". I would suggest to be more specific (here or in method): How many experts? What is their level of experience? Did they annotations overlap and if so, was variability assessed?
- * "photographed at 400x original magnification". The resulting pixel size should be given to be able to compare with scanned images available on other databases for example.
- * the authors say the images were split into training/validation/test set. If it is the case and one patient has several images, then, it is possible that a patient has some of his slides in the training set, and some in the test set? The cross-validation accuracy (which is the one that matters) shows the accuracy drops from 99.7-100% to 80-82%, which is a big drop. It either suggests that the slides are very different, or that there is a patient leak when tested on the same hospital's dataset. Please double check and show which case it is. One way to un-ambiguously prove the sample preparation is the reason for that drop would be to (re)strain slides from a subset of patients using the procedure of the different hospitals. If not, showing the list of patients in each dataset is different is another option.
- * Regarding the validation, it looks like a bunch of parameters were tested and optimized. It could be interesting, in a supp table, to show how each parameter tested affected the validation performances and led to the selected optimal conditions.
- * Authors use "accuracy" only to assess the performance. Other measures are usually more common and could be considered (AUC for example).
- * What is the ground truth /gold standard used to assign the true label to each patient? If it was based on hematopathologists diagnosis, please how differences between pathologists were tackled to assign ground truth.
- * How are the outputs of all the trained networks combined to generate the final decision score?
- * Which network(s) contribute the most/the best to the output, from a statistical point of view, and from "workbench" tests? It would be good to show the performance of each individual network to solve this task to prove the benefit of combining the 17 of them.
- * The different networks used deal with different input sizes. How did the authors deal with it?
- * The links to the image and code are not yet working. It is not possible to assess the feasibility of the technique without having a view on the code. Furthermore, as often requested by most journals, code must be properly commented and written in a way that will allow users to reproduce the results.

As a conclusion, I think the main interesting point of this paper is the fact that they combine 17 different neural networks to achieve the classification. However, the lack of precision in the method, the lack of intermediate results and lack of analysis of the obtained results cast a significant shadow on the current manuscript. I hope the authors will solve the concerns and questions addressed because if they show there is no patient leak and the 17-network combination performs significantly

better than each individual ones, then it would be a very interesting paper.

Point-by-point response to reviewers

Reviewer #1 (Remarks to the Author):

Li et al are presenting a deep learning platform including multiple convolutional neural networks for the diagnosis of human diffuse large B cell lymphoma (DLBCL) from histopathology slides of the lymph. Using data from three different hospitals, their method achieved diagnostic rate of 100% for hospital A, 99.71% for hospital B and 100% for hospital C respectively. However, none of the three models is generalizable, which significantly limits the impact and novelty of this work. Previous studies have shown that deep learning models for histologic diagnosis can be generalized in several cancer types.

If we may, we respectfully disagree with the reviewer's comment on the generalization performance of our deep learning models based on the following facts:

a. We guess that this comment by the reviewer is derived from our results shown in Figure 3 in our originally-submitted manuscript, indicating roughly a 20% drop of DLBCL diagnostic accuracy when we conducted a cross-hospital model testing. The reviewer mentioned *“Previous studies have shown that deep learning models for histologic diagnosis can be generalized in several cancer types”*. Although we don't know which study the reviewer was referring to and what degree of generalization was shown in that study, based on our knowledge, we would like to point out that a drop of diagnostic accuracy or sensitivity in a generalization test has been observed in every published work. This is simply because the technical variability introduced mainly by tissue slide preparation and the types of imaging equipment used significantly affect the performance of any given deep learning model and each hospital or institution determines its own way of preparing the tissue slides and what imaging equipment to use. The technical variability among different hospitals/institutions would not be eliminated until one day all hospitals/institutions conduct standardized procedures for tissue slide preparation and image collection. However, the technical variability could be largely eliminated within one hospital or institution, which was what we did in our study.

To further emphasize our point regarding the generalization of our models, please allow us to discuss, in detail, a beautiful representative work published recently by a research group in Memorial Sloan Kettering Cancer Center (Campanella, G. *et al.* Clinical-grade computational pathology using weakly supervised deep learning on whole slide images. *Nat Med* 25, 1301-1309, 2019), which was actually cited in our originally-submitted manuscript. In this study using AUC (area under the curve) as an indicator for diagnostic sensitivity of solid tumors, the highest sensitivity reached 0.98 by the deep learning model established. Importantly, when the model was used to test a new dataset, a 20% drop of diagnostic sensitivity was observed, which represents the current level of deep learning models in computational pathology and is similar to the generalization level we observed for our models in our cross-hospital DLBCL study. On the other hand, we believe that our model should perform better in generalization than the models published by others, because our model was built on more advanced deep learning methods (see next section below in “b”).

b. We should thank the reviewer for questioning the generalization ability of our deep learning models, because it prompted us to ask why our model did not show a higher level

of generalization compared to the published work (Campanella, G. *et al.* Clinical-grade computational pathology using weakly supervised deep learning on whole slide images. *Nat Med* 25, 1301-1309, 2019). We ask this question to ourselves because we are the first to be able to combine 17 CNNs and use our unique Global Optimization Method to read pathologic slides and we should do much better. By thinking so, we suddenly recognized one “error” we made when we conduct the two cross-hospital tests of our models.

In the first test when using the model A established in hospital A to read the slide images of patients from hospital C, which showed a 82.09% diagnostic accuracy, we did not standardize the shape of the tissue images when conducting our generalization test between the two hospitals. The model A was trained with the original rectangular images (the width to height ratio is 4:3) from the hospital A and the tested images from hospital C were squares in shape; thus, the images from hospital C were twisted and were fed into the model A for testing. Fairly, we had a reason for not unifying the shape of the images because we did not have a need to do that when we focused more on building hospital-specific models as described in our originally-submitted manuscript. Now, we realize that our generalization test involved two different hospitals, which definitely requires reading the same shape images for precision. This is because the variability caused by the differences in the shape of the images are totally unrelated to pathological appearance of the disease but would be picked up by the deep learning model, which could explain why the performance of our model in the generalization test is expected to be better but was at a level similar to what was observed by others. Therefore, we re-did this generalization test by normalizing the shape of 179 images in hospital C to the shape of the images in hospital A to determine the generalization ability of our model trained and built using the images in hospital A. The result showed that the diagnostic accuracy in this new cross-hospital test was significantly increased from 82.09% to 90.50% (see new Fig. 3c in the revised manuscript; for the reviewer’s convenience, we showed this new subfigure here as well). This

Fig. 3c. A high diagnostic accuracy of our AI models. Hematoxylin and eosin (H&E)-stained formalin-fixed paraffin-embedded tissue sections prepared from lymph nodes of DLBCL and non-DLBCL patients from four unrelated hospitals (Hospitals A, B, C, D) were photographed (Hospitals A, C) or scanned (Hospitals B, D) at 400x original magnification to produce pathologic images for generating three separate AI models (Models A, B, C), each of which was specifically generated using the DLBCL and non-DLBCL samples from the corresponding hospital. A high diagnostic accuracy was reached by the three AI models (100% for Hospital A, 99.71% for Hospital B, 100% for Hospital C, respectively (see Fig. 3a in the manuscript). However, the diagnostic accuracy dropped from 100% to 90.50% or 82.09% with or without unifying the size and shape of the images between hospital A and hospital C when cross-hospital use of the deep learning model A was used to read the slide images of patients from hospital C. The diagnostic accuracy increased to 100% when the model A was used to read new images of patients in the same hospital. 100% diagnostic accuracy was also achieved when the model B was used to reach the slide images of patients from a new hospital (hospital D) after elimination of the technical variability introduced by slide preparation procedures and image collection equipment.

result demonstrates that our deep learning model has reached a high level of generalization compared to the methods established by others (Campanella, G. *et al.* Clinical-grade computational pathology using weakly supervised deep learning on whole slide images. *Nat Med* 25, 1301-1309, 2019). We should mention that in this generalization test, we used the model A established in hospital A to read the images from hospital C, because the images from both hospitals were prepared by photographing.

In the second test when using the model B established in hospital B to read the slide images of patients from hospital A, which showed a 79.80% diagnostic accuracy, was that we intentionally ignored the fact that the images from hospital B were collected using a scanner, whereas the images from hospital A were collected using a camera. Thus, it was inappropriate and not meaningful to conduct the generalization test between hospital B and hospital A, and an accurate assessment of the generalization ability of our model would be to scan the tissue slides in both hospitals (see “c” below).

c. Although the 90.50% diagnostic accuracy from our new generalization test described above is among the best, we wonder what would be the other factors that prevented us from reaching an accuracy close to 100% in the cross-hospital (generalization) test as we achieved in our accuracy tests within each hospital. The published work (Campanella, G. *et al.* Clinical-grade computational pathology using weakly supervised deep learning on whole slide images. *Nat Med* 25, 1301-1309, 2019) has shown that the technical variability can be introduced by slide preparation methods and imaging equipment used in different institutions. In other words, slide preparation procedures and types of scanner or camera used could negatively affect the performance of deep learning models. Specifically, in this published study, about 3% drop in diagnostic sensitivity was observed when different scanners were used between institutions. In addition, the difference in slide preparation caused about 6% drop in diagnostic sensitivity. Thus, the technical variability introduced by slide preparation methods and imaging equipment caused about 9% drop in diagnostic sensitivity. If we could eliminate the interference by those two factors (slide preparation methods and imaging equipment), our deep learning model from hospital A (model A) would reach, when reading the images from hospital C, a diagnostic accuracy close to 100% (from 90.50%) for DLBCL. In our view, however, it would be difficult to eliminate those two factors through improving deep learning models, but a practical way to do is to standardize slide preparation procedures and image collection methods among different hospitals, which would require taking an international cooperation effort in the computational pathology field. Achieving this cooperation in a large scale is unrealistic at present, although we believe it will eventually happen. In contrast, when we focused on each individual hospital, we could largely eliminate those two factors, because we found that in each hospital, all tissue slides (DLBCL and non-DLBCL) were prepared similarly and pathologic images were collected using the same equipment (a scanner or camera). This explains why our hospital-specific deep learning models reached 100% diagnostic accuracy for diagnosing DLBCL. This also explains why we emphasized building individual hospital-based deep learning models for achieving 100% diagnostic accuracy, which obviously requires an ability to generate deep learning models using a smaller dataset as we showed in our manuscript. We believe that at present, this strategy is practical, bringing us closer to using our deep learning models for DLBCL diagnosis in medical practice. To further demonstrate the practicality of this strategy, we conducted a “generalization-like” test, in which we used the model A established in hospital A to read new images (totally 110 images) that were obtained from the same hospital but not used for training the deep learning model (model A). We thought that this would largely eliminate those two factors (slide

preparation methods and imaging equipment) that caused about 9% reduction in accuracy between two different institutions (Campanella, G. *et al.* Clinical-grade computational pathology using weakly supervised deep learning on whole slide images. *Nat Med* 25, 1301-1309, 2019). Our new result showed that diagnostic accuracy remained at 100% (see new Fig. 3c in the revised manuscript and also the figure above in this rebuttal letter), suggesting that we could eliminate the technical variability introduced by slide preparation and imaging equipment through building a single hospital-based deep learning model to achieve a 100% diagnostic accuracy. Importantly, this result also suggests that our deep learning models with the 100% accuracy did not pick up signals unrelated to DLBCL because we used, as a control, non-DLBCL tissue images that were prepared and collected similarly within the same hospital. In other words, the overfitting is not an issue of concern in our study.

To further evaluate the generalization ability of our deep learning model, we conducted a new cross-hospital test, in which we used our model established in hospital B (model B) to read new tissue images of patients from a different hospital with a goal of eliminating the technical variability introduced by slide preparation procedures and image collection equipment. Specifically, we reached out a new hospital (hospital D) that was not originally involved in our model building when we submitted our original manuscript reviewed by the reviewer, to obtain DLBCL and non-DLBCL samples, because this hospital utilized the slide preparation procedures similar to the ones used by hospital B. Also, we scanned the pathologic slides from hospital D to produce whole slide images (totally 135 images) using the same scanner we used to collect whole slide images from hospital B. Thus, we can say that we basically eliminated the differences between the two hospitals (hospitals B and D) in slide preparation and image collection equipment. We then used the model B established in hospital B to read the images from hospital D, and 100% diagnostic accuracy for DLBCL was achieved in this cross-hospital test.

Taken together with all new tests we did in revising our work, we think it would be fair to say that we have generated accurate deep learning models with high generalization ability for diagnosis of DLBCL.

Major Comments

1. Combining multiple classifiers is not a novel idea, and also the authors need to compare the performance of their approach to each individual CNN. Also, how robust are the results to the data split (training, validation, testing)?

We respectfully argue that our ability to combine 17 CNNs and use them together as a single model is definitely unprecedented. This single model has all of the layers built in those 17 CNNs for conducting transfer deep learning with our datasets, and this novel approach allowed us to achieve a high diagnostic accuracy for DLBCL.

We should point out that different research groups, including us, have used different CNNs individually but diagnostic accuracy has not been satisfactory. In our study, we initially used each of the 17 CNNs, respectively, to analyze the pathologic images of DLBCL in each of the three hospitals, and found that the average percentage of diagnostic accuracy in the three hospitals by using one CNN was ranged from 87% to 96%. In our view, the diagnostic accuracy needs to be 100% or greater 99% prior to employing any deep learning model in medical practice. This is why we programmed multiple models (17 CNNs) into one system with our algorithms to

enhance the performance of deep learning with a goal of achieving 100% diagnostic accuracy. As a result, we have indeed reached 100% accuracy, which is superb to a sole use of any one of the 17 CNNs. In our originally-submitted manuscript, we did not mention this information regarding the comparison between the combined use of all 17 CNNs and the sole use of one of the 17 CNNs, because we thought we did not have a need to do so. In this revised manuscript, we added this information, as the reviewer requested, to help to explain why we combined the 17 CNNs in our analysis of the DLBCL tissue images.

In terms of the robustness of our results to data split, we can say that the results of our model are very robust to the data split. We used 0.8-0.1-0.1 (training 0.8, Validation 0.1, Testing 0.1) during the training and also tried other ratios such as 0.6-0.2-0.2 and 0.7-0.15-0.15, but we did not notice significant differences, which indicates that the data split ratio in training does not play a significant role in our deep learning platform.

2. The authors can build a model using the data from Hospitals A and C together since the images are obtained with the same method. This could address the sample variation since now the model has taken the variations into account during training. This could potentially yield a more generalizable model.

Here the reviewer emphasized again the importance of having a more generalizable model that could be built in one hospital and used in a different hospital. As we addressed above the similar concern by the reviewer, the most difficult obstacle to building a more generalizable model is the technical variability introduced by slide preparation methods and imaging equipment used in different institutions, which would be difficult to overcome by further improving deep learning models. In other words, the technical variability produced among different institutions would not be effectively eliminated until all institutions conduct standardized procedures for tissue slide preparation and image collection. Although the tissue images from hospitals A and C were similarly collected by photographing, we knew that the differences still existed in tissue preparation procedures and types of imaging camera used, both of which were out of our control. Thus, combining data from Hospitals A and C in model generation would not substantially improve the diagnostic accuracy when using the model (model A+C) to read slides from a third hospital in which slide preparation method and imaging equipment are likely different from those in hospitals A and C, again because the third hospital would prepare tissue slides differently and use a different imaging device compared to hospitals A and C. Regardless, it is fair and appropriate for the reviewer to question our deep learning model for its ability to provide high diagnostic accuracy when analyzing new pathologic images that are not used in training the model. As described above in our response to the first concern by the reviewer, we thought that the best way for conducting our generalization test is to use our deep learning model established in hospital A (model A) to read the images from hospital C. Actually, we did this test and showed in our originally-submitted manuscript that a 20% drop in diagnostic accuracy was observed, which is comparable to the 20% drop of diagnostic sensitivity in the published study on diagnosis of solid tumor using deep learning model (Campanella, G. *et al.* Clinical-grade computational pathology using weakly supervised deep learning on whole slide images. *Nat Med* 25, 1301-1309, 2019). We questioned ourselves by asking why our deep learning model (model A) did not show a less drop in diagnostic accuracy compared to available work by others. As described above in our response to the first concern by the reviewer, we discovered some differences in the shape of the images between hospital A and hospital C, which could severely affect the diagnostic accuracy. We corrected this mismatching by normalizing

the shape of 179 images in hospital C to those in hospital A and used the model A to read these normalized images in hospital C. As a result, our cross-hospital diagnostic accuracy was significantly increased from 82.09% to 90.50% (see new Fig. 3c in the revised manuscript and Figure 3 shown in this rebuttal letter above), indicating that our deep learning model is actually more generalizable compared to the models established by others. Because the technical variability introduced by slide preparation methods and imaging equipment caused about 9% drop in diagnostic sensitivity (Campanella, G. *et al.* Clinical-grade computational pathology using weakly supervised deep learning on whole slide images. *Nat Med* 25, 1301-1309, 2019), our model would reach a diagnostic accuracy close to 100% for DLBCL, after elimination of this technical variability by standardizing slide preparation and image collection. As described above in response to a similar concern by the reviewer, we validated this assumption by using the model B established in hospital B to read pathologic images of patients from a different hospital (hospital D) where the slide preparation procedures were similar to the ones used in hospital B. Also, the same scanner was used to collect whole slide images from both hospital B and hospital D. Together, we basically eliminated the differences between the two hospitals (hospitals B and D) in slide preparation and image collection equipment. We achieved 100% diagnostic accuracy for DLBCL in this cross-hospital test (see new Fig. 3c in the revised manuscript and Figure 3 shown in this rebuttal letter above).

To further test the ability of our deep learning model in reading new DLBCL images, as described above in our response to the first concern by the reviewer, we took another approach by using the model A established in hospital A to read new images (DLBCL or non-DLBCL) that were obtained from the same hospital but not used for training the deep learning model (model A). The result showed that diagnostic accuracy remained at 100% (see new Fig. 3c in the revised manuscript and also Figure 3 shown above in this rebuttal letter), demonstrating that our model is highly generalizable in reading new images. This result also suggest that we could eliminate the technical variability introduced by slide preparation and imaging equipment, prior to achieving cross-hospital standardization of slide preparation and imaging collection, through building a single hospital-based deep learning model and achieving a 100% diagnostic accuracy.

3. Some of the data, as stated in line 98, were obtained by taking photos of slides and some by scanning them. Is there any difference at the resolution? At which resolution where the whole slides from Hospital B scanned?

There is a difference in the imaging resolution between photographing and scanning, because we just used whatever was available in each hospital. For the scanned whole slide images from hospitals B and D, the resolution is 0.121547 (micrometer/pixel) and the image size obtained for training and testing is 945x945 pixels.

4. The order of presenting the results is a little confusing. It is not clear that the authors built 3 different models (one for each hospital) because of the difference in the sample preparation and collection (microscope camera vs scanned) until the paragraph “Effect of sample preparation procedures on diagnostic accuracy”. It will be better to re-write/merge this paragraph with the previous one “Achievement of a high diagnostic accuracy for clinical use” and maybe the next one “Establishment of AI models using small datasets” to improve the flow of the paper.

We apologize for the confusion and appreciate very much the reviewer's suggestion that we should re-write/merge those two paragraphs. Following the reviewer's suggestion, we made the changes in this revised manuscript.

5. Regarding the whole slide model, it seems that the authors used parts of the image to build their model (line 176). Since the CNNs mentioned in Fig 1 have the capacity to tile the images why wasn't the whole image used?

Like many other previous studies, we, if needed, would be able to tile the images and get each every piece of tiled images classified in building our model. However, we intended to save computing resources by randomly selecting nine pieces of images within the DLBCL tissue area clearly annotated by pathologists for each case. In perspective, the majority of pathology departments in hospitals do not have access to and the training to use high-performance super computers to analyze entire tissue image of each case, which is one of the reasons why we used parts of the image to hope to establish an AI system that could be used by all hospitals.

6. To better support the claim in line 183 about the "unprecedented" high diagnostic accuracy of their models, it is important that the authors address fully in their introduction the performance of the current and other published methods to diagnose DLBCL.

We totally agree with this suggestion by the reviewer. We could only find one DLBCL-related study showing a 95% diagnostic accuracy along with significant false-negative and false-positive rates (Achi et al. Annals of Clinical & Laboratory Science. vol. 49, no. 2, 2019). By our standard, it is impossible to translate this kind of results into clinical use. By contrast, our deep learning models achieved a 100% diagnostic accuracy without having false-negative and false-positive cases. We have included this reference under the "Discussion" section in our revised manuscript.

7. The CIFAR-10 dataset is a collection of slides with labels such as "airplanes, cars etc". The GOTDP-MP-CNN platform generated by the authors outperformed all other architectures included in the CIFAR-10 platform in classifying these images. I am not sure if this comparison is relevant to DLBCL diagnosis.

The goal of this analysis of the CIFAR-10 dataset was to show that our deep learning algorithm among or above the best available ones, which helps to explain why we were able to achieve a 100% diagnostic accuracy for DLBCL. If the reviewer strongly suggests that we should remove this information, we would do that, although we ask to still include this result as supplemental information.

8. The comparison with the pathologists' performance is interesting and indeed it is demonstrating the network's high performance. Is looking at 400x images only the common practice for pathologists to make a diagnosis? Do they also look at IHC results? I want to make sure that the network's performance is compared to the actual state-of-the-art practice in the clinic. Also, are the 7 pathologists experienced in DLBCL diagnosis?

This is a great question! It is great because it gives us an opportunity to make one point: for the same tissue slide, there is a fundamental difference in the way of reading the slide by artificial intelligence (AI) vs. a pathologist. Unlike what a pathologist does, AI does not use the concepts for cell membrane, cytoplasm or nucleus, etc., and it picks up signals that are totally different from those looked at by the pathologist. Also, one of the major goals of using AI in

computational pathology is to reduce the workload of pathologists. Therefore, AI should not be expected to follow and repeat what pathologists do in their daily practice. We believe that the principal of employing AI in computational pathology should be to collect and read tissue images as less as possible but can still achieve high diagnostic accuracy. It is true that in common practice by pathologists for DLBCL diagnosis, at least they look at both 200x and 400x images as well as IHC results, but this is because they could not make a correct diagnosis by only looking at a 400X image; they often need help from looking at the multiple images from the same patient by multiple ways (morphology of the cells, biochemical changes detected by immunohistochemistry, etc.). In contrast, our AI model only needs to read 400x images to achieve a 100% diagnostic accuracy for DLBCL. Thus, it is obvious that AI will help to reduce workload of pathologists in a big way.

The 7 pathologists have different levels of experience in diagnosing DLBCL, and as what we showed, none of the 7 pathologists was 100% correct in the diagnosis of DLBCL when they only looked at 400x images, which reflected the fact that in medical practice, pathologists need to look at multiple images of each patient by multiple ways to make correct diagnosis. This further emphasizes the need of AI in assisting pathologists in their daily practice. On the other hand, a significant number of hospitals in the world do not have “*state-of-the-art practice in the clinic*”, which is labor-intensive and expensive, and again, AI will definitely help to reduce the labor and cost.

9. In line 277 is the first time we see the term “GOTDP-MP-CNN” used in the paper although it appears in Fig1 without explanation except for the legend. It would be helpful if the term appeared earlier in the manuscript.

We apologize for the inconvenience and have made the correction in our revised manuscript.

10. Will the images be made available to the scientific community?

Definitely yes.

Minor Comments

1. Regarding the title of the study, it might be a little redundant to have both terms “artificial intelligence” and “deep learning”.

Totally agree. We removed artificial intelligence from the title in our revised manuscript.

2. In line 75, the word “Acquisition” should probably be lowercase.

The reviewer’s correct, and we changed it to lowercase in our revised manuscript.

3. In Fig1b, would be more aesthetically pleasing to align all the squares properly.

We made this change in our revised manuscript.

4. In Fig3 I don’t see a legend for panel b.

We apologize for this mistake. We added the legend for panel b of Fig. 3 in our revised manuscript.

5. Line 188: probably meant to say “In contrast, ”

We apologize for this mistake, and it should mean “In contrast”. We corrected this error in our revised manuscript.

6. Fig 4a. The little yellow box is unreadable and may need to be removed or enlarged.

Agree. We removed it in our revised manuscript.

Reviewer #2 (Remarks to the Author):

I am not expert in AI systema as I am an haemopathologist, therefore my objections are based only on the histopathology side of the story. I accept the AI can be great, I have no way to judge that but I am worried as far as the handling of the pathology side is concerned.

We thank the reviewer for supporting our study by saying: “*I accept the AI can be great...*”.

I have several worries.

Against what the authors measure the accuracy? How were diagnosed the cases used as DLBCL or not-DLBCL to train the system? Which immunohistochemical panels were performed? Which was the anatomical location of the samples: only lymph nodes or also other organs?

These are great questions and the answers to them were actually described in detail under the “Methods” section in our originally-submitted manuscript. We apologize for not being able to provide sufficient information under the “results” section due to the format requirements by the journal.

To answer the “accuracy” question by the reviewer, please allow us to introduce again the following definitions we used in our manuscript:

DLBCL: positive for DLBCL

Non-DLBCL: negative for DLBCL (healthy or other diseases)

True positive (TP): the number of cases correctly identified as DLBCL

False positive (FP): the number of cases incorrectly identified as DLBCL

True negative (TN): the number of cases correctly identified as healthy or other diseases

False negative (FN): the number of cases incorrectly identified as healthy or other diseases

The accuracy of a test is its ability to differentiate the DLBCL and Non-DLBCL cases correctly. To assess the accuracy of a test, we calculated the proportion of true positive and true negative in all evaluated cases. Mathematically, accuracy = $(TP+TN)/(TP+TN+FP+FN)$. In other words, the accuracy reflects the correct rate of diagnosis. For example, if we used our deep learning model to read slide images for 100 cases of DLBCL (25 cases) and non-DLBCL (75 cases) and identified 24 cases of DLBCL and 75 cases of non-DLBCL correctly but missed 1 case of DLBCL, TP = 24, TN = 75, FP = 0, and FN = 1. Diagnostic accuracy for DLBCL is $24+75/24+75+0+1 = 99\%$.

Every case of DLBCL or non-DLBCL was correctly diagnosed based on reading tissue slides with H&E staining and immunohistochemical detection of expression of B cell markers such as CD20 and PAX5 by pathologists. The diagnosis and classification of DLBCL, and of non-DLBCL lymphomas in the control group, was made in accordance with the WHO classification of tumours of haematopoietic and lymphoid tissue. The diagnosis and classification

of DLBCL, and of non-DLBCL lymphomas in the control group, was made in accordance with the WHO classification of tumours of haematopoietic and lymphoid tissue. Also, the pathological diagnosis of each case was consistent with clinical symptoms of the patient.

For the anatomical location of our samples, only lymph nodes were used, which we describe more clearly in our revised manuscript.

DLBCL is an heterogeneous group of lymphomas. What about the T cell rich B cell lymphomas, the Burkitt's lymphoma , the high grade B cell lymphoma, the double hit DLBCL and the post transplant DLBCL, just to mention the main types.

The reviewer is absolutely correct: DLBCL is a heterogeneous group of lymphomas. In our study, the DLBCL cases were limited to those subclassified as DLBCL, NOS according to WHO classification, and realistically we did not set a goal for us to develop an AI pathology system for many subtypes of human lymphomas within one study. We know that in daily practice of pathologists, there is a need for DLBCL diagnosis with exclusion of other B-cell lymphomas with large-cell morphology, including mantle cell lymphoma, lymphoblastic lymphoma, and plasmablastic lymphoma, etc. It is also necessary to exclude malignant tumors of other histogenesis including carcinoma, melanoma, and sarcoma that may potentially mimic DLBCL. However, in our current work, we focused on distinguish DLBCL from non-DLBCL to prove the principal that our deep learning models are accurate and efficient in diagnosis of DLBCL. Our next goal in the future will be to detail with the heterogeneity of DLBCL to subclassify the disease via deep learning.

Was the non-DLBCL group composed by the same diagnosis in all the hospital?

Among the hospitals involved in our study, the non-DLBCL cases in each hospital included similar medical conditions or diseases, some of which need to be distinguished from DLBCL in diagnosis as we explained in our answer to the previous question by the reviewer.

The authors do not provide a list of these diagnosis of the cases used as non-DLBCL. This is a critical omission as the composition of this list is critical to properly train the system.

We would like to mention that when building our deep learning models, we purposely did not want to know the specific diagnosis for each non-DLBCL case because we intended to eliminate any possible human influence on our model building. On the other hand, when obtaining pathologic slides from different hospitals, we were bound to policy restrictions by relevant institutional committees, and often we are not allowed to have a detailed list showing which patient has what disease. However, what we did require to know from the four hospitals involved in our study is that each non-DLBCL case is indeed not DLBCL. In addition, for the non-DLBCL cases in our study, we purposely included some cases with pathological appearances that are similar to DLBCL and clinically need to be distinguished from DLBCL. Following the reviewer's suggestion, in our revised manuscript, we listed the types of disease for non-DLBCL, which included: benign/reactive lymph nodes, metastatic carcinomas, small lymphocytic lymphoma/chronic lymphocytic leukemia, mantle cell lymphoma, follicular lymphoma, and classical Hodgkin lymphoma.

Variation in quality of histological preparation among hospital is a very well-known problem.whether. Ther authors could have started to study it, for example by repeating the

analysis after cutting and staining all the case just in one hospital: this simple measure could already improve diagnosis across hospitals. Another cause could have been the different composition of the non DLBCL list in different hospitals.

We appreciate that the reviewer agrees with us regarding the problem in tissue slide preparation. As we described in response to the reviewer 1 above, we proposed two ways to overcome this problem: 1) standardization of slide preparation procedures and tissue image collection equipment, which would take international effort across hospitals; and 2) establishment of a single hospital-based deep learning model to achieve a high diagnostic accuracy, which would require having an ability to build the model using a smaller dataset, as we achieved in our study. At present, the second strategy is more realistic and practical, we believe.

The reviewer has suggested that we could cut and stain all cases (from all hospitals involved in our study) in just one hospital to improve diagnosis across hospitals, and we believe that doing so would increase diagnostic accuracy across these hospitals. However, we have to respectfully argue that the reviewer's suggestion is unrealistic to us simply due to the legal and ethical issues related to intellectual property, institutional review boards (IRB) approval under FDA regulations, patient right protection, etc. In other words, it would be extremely difficult to take formalin fixed paraffin embedded tissue blocks out of any hospitals to process them somewhere else without violation of institutional rules and regulations. This is why we proposed and emphasized in our study that at present, a realistic and practical way to do is to focus on each individual hospital when building a deep learning model within that hospital, which will also largely eliminate the technical variation introduced by histological slide preparation and image collection. As we mentioned in response to the reviewer 1 above, the published work showed that the technical variability introduced by slide preparation and image collection caused about 9% drop in diagnostic sensitivity of solid tumors between hospitals (Campanella, G. *et al.* Clinical-grade computational pathology using weakly supervised deep learning on whole slide images. *Nat Med* 25, 1301-1309, 2019). This 9% drop is consistent with the percent drop in diagnostic accuracy of DLBCL in our study when we utilized the model A (established in hospital A) to read tissue images from hospital C after normalizing the shape of the images of patients from both hospitals: dropping from 100% accuracy to 90.50% (see new Fig. 3c in our revised manuscript and above in this rebuttal letter in response to the reviewer 1). In this new figure, we also showed that when we utilized the model A to read new tissue images from the same hospital (hospital A), we maintained 100% diagnostic accuracy of DLBCL, indicating that the drop from 100% to 90.50% when utilizing the model A to read the images from hospital C was indeed caused by the technical variability (slide preparation and image collection) between the two hospitals. This conclusion was further supported by our achieving 100% diagnostic accuracy when we used the model B established in hospital B to read the images from a different hospital (hospital D) (see new Fig. 3c in our revised manuscript and above in this rebuttal letter in response to the reviewer 1), which is because the slide preparation procedures are similar between hospital B and hospital D, and the same scanner was used to collect the images for those two hospitals. Again, we believe that the best way to eliminate this technical variability is to standardize slide preparation procedures and tissue image collection equipment, which would take international effort across hospitals.

It is reasonable and legitimate to think that **“the different composition of the non DLBCL list in different hospitals”** may have contributed to the drop in diagnostic accuracy, but we believe that it played, if any, a much lesser role, because if it played a larger role, we would not be able to achieve the high diagnostic accuracy in every individual hospital involved

in this study, especially when we used the model B established in hospital B to read the images from a different hospital (hospital D) and achieved 100% diagnostic accuracy in this cross-hospital test. On the other hand, from our experience we believe that there are many fundamental pathological differences between DLBCL and non-DLBCL and our AI models can pick them up in a way that is totally different from a human pathologist does. Furthermore, with respect to our AI models, **“the different composition of the non DLBCL list in different hospitals”** would not be a concern because non-DLBCL do not have some unique pathological features of DLBCL. In other words, our AI models are capable of finding these pathological features of DLBCL to determine which cell is DLBCL or non-DLBCL, regardless which hospital the non-DLBCL cases are from simply because non-DLBCL does not have the unique pathological features of DLBCL. Again, the technical variability introduced by slide preparation and image collection equipment across hospitals caused about 10% drop in diagnostic accuracy, and we believe that as DLBCL did, non-DLBCL also contributed to this technical variability. The technical variability can be basically eliminated by building hospital-specific deep learning models as we successfully did in our study because DLBCL and non-DLBCL slides can be handled in the same way within the same hospital. Alternatively, the technical variability can be largely eliminated by standardizing slide preparation procedures and tissue image collection equipment in all hospitals.

This variability makes this system of scarce practical utility as hemopathology is usually performed in centralised referral centres that cover many different hospitals.

Actually, most decent sized academic hospitals in the US, which are located in most states, employ hematopathologists and are able to work up and diagnose such cases without referral to a specialized center. It is true that community hospitals or private groups of pathologists may need to send these to a more specialized center. Nevertheless, this technical variability introduced by slide preparation and image collection equipment would be eliminated if all tissue samples from different hospitals were sent to the same **“centralised referral centres”** where the same slide preparation procedures and tissue image collection equipment are employed. However, the technical variation across different centers still exist and would negatively affect the diagnostic accuracy if a deep learning model established in one center were used to read slide images from a different center. The solution would be: 1) to send all tissue samples from all hospitals in the world to one referral center, which is apparently unrealistic; 2) to standardize slide preparation procedures and tissue image collection equipment in all referral centers in the world, which is obviously not possible at present; and 3) to build referral center-specific deep learning model to read tissue images of all patient samples sent to a particular referral center from **“many different hospitals”**. In other words, a deep learning model established in a particular referral center will only be used to read tissue images of patient samples to be sent to this referral center from different hospitals. Thus, we respectfully argue that utility of our deep learning models within the hospitals where the models are built will be practical and helpful for pathological diagnoses of human diseases including DLBCL, in terms of reducing labor and increasing diagnostic efficiency and accuracy by pathologists.

To investigate the feasibility of obtaining an AI system in each hospital the authors worked on a set of 24 DLBCL patients and 35 non-DLBCL individuals. I am an hemopathologist, not a statistician or a computer scientist, however a set of 35 patients other than DLBCL does not even cover the number of entities other than DLBCL normally seen by an haemopathologist.

Actually, in hospital A, we had 500 cases of DLBCL and 505 cases of non-DLBCL, which covered many other diagnoses. On the other hand, in our current study, we did not intend

to cover AI diagnosis of all types of human hematopoietic diseases, rather we focused on a common form of human lymphoma DLBCL to begin to develop deep learning models for diagnosis of DLBCL and in the future for all other human diseases whose diagnosis depends on pathological analysis of diseased tissue. Our final goal is to build disease-specific deep learning models and use them as a whole to help to diagnose all these human diseases. We, as a scientific community, have to start this long journey somewhere and wish that the reviewer would support our effort in helping pathologists to utilize the AI system in pathological diagnoses of human diseases, starting from DLBCL and expanding to all other diseases eventually.

When the author discuss the detection of a false negative diagnosis and review the case, it is unclear why they decided that the case was not a DLBCL. They do not provide enough information, in particular immune profile, proliferating fraction and proliferating fraction distribution in order to properly conclude that the case was not a DLBCL. The diagnosis of “follicular diffuse mixed type” (probably what Lennert called “diffuse centrocytic centroblastic”) is now mostly recognised to be DLBCL. Alternatively it can represent the extrafollicular component of a follicular lymphoma but for a pathologist to make a mistake is pretty bad!

We do not know the detailed information about any particular patient due to confidentiality agreements with hospitals, but what we do know is that if a case is labelled as non-DLBCL, which was based on pathological appearance recognized by pathologists, immunohistochemistry, molecular biology, clinical symptoms, etc., this case is definitely not a DLBCL. During our study, tissue slides were re-examined and confirmed by at least one experienced pathologist to ensure the correctness of diagnosis for all patients included in our study.

We appreciate that the reviewer pointed out a real difficult situation where a pathologist feels it could be challenging to distinguish DLBCL from some related human conditions such as “*follicular diffuse mixed type*”, *follicular lymphoma*”, etc., as we know that some form of lymphoma such as follicular lymphoma can transform to become DLBCL at a later stage of the disease. This is a good example for what a deep learning model can do to help pathologists: not only reducing their labor in reading tissue slides but also assisting them to correctly diagnosis DLBCL and other human diseases. If we may, we would like to mention again that pathologists should not think that AI and pathologists see a tissue slide in a similar way, and in fact, in totally different ways. This is one dilemma for pathologists to overcome because they often use what pathologists can see to evaluate what AI should and can do. The reality is that AI has surpassed human performance in many areas, including, for example, playing chess (Silver, D. *et al.* Mastering the game of Go with deep neural networks and tree search. *Nature* **529**, 484-489, doi:10.1038/nature16961, 2016). Actually, we, human beings, do not know how a machine with a deep learning ability thinks during the chess game, which is why the best chess player was defeated by the machine, as the chess player thinks like us but the machine thinks differently. Anyway, we should believe that AI can help pathologists to do many things and do better under the supervision by pathologists.

The authors write:

“To be careful, we compared the diagnostic accuracy of our AI model with that of seven invited pathologists for reading the same set of DLBCL and non-DLBCL 400x images. The pathologists used about 60 minutes in average to read all 531 images, and the highest

diagnostic accuracy among the seven pathologists was 74.39% (Fig. 4b). In contrast, it took less than 1 minute with a notebook computer for our AI model to finish reading all these images, and the diagnostic accuracy was 100%.”

I guess they mean that the pathologists only used an HE. But I hope that the AI system was trained using diagnosis done with the support of immunohistochemistry. This experiment to me make no much sense. The gold standard is to make the DLBCL diagnosis with immunohistochemistry, not only to verify the B nature of the malignancy but also to establish subtypes (like e.g. Germinal Centre vs non-Germinal Centre type, Burkitts; expression of cMyc).

Furthermore to give 60 minutes to look at 531 cases is non sensical: of course at that speed an human pathologist will fail, is a self-fulfilling prophecy. Plus it is misleading to test the system against diagnosis done only on HE: this is not the normal clinical practice.

We thank the reviewer for asking these clinical-relevant questions, and please allow us to answer them one-by-one below.

It is true that the pathologists we invited only looked at the HE tissue slides, which was why they did not achieved 100% diagnostic accuracy as our AI model did. This fact does not mean that these pathologists were incapable of providing correct diagnosis, rather it emphasizes that pathologists often need support from other clinical tests (such as immunohistochemistry, molecular biology, etc.) to come up with a correct diagnosis, as the reviewer pointed out that *“The gold standard is to make the DLBCL diagnosis with immunohistochemistry, not only to verify the B nature of the malignancy but also to establish subtypes...”*. We totally agree with the reviewer that the gold standard is to make the DLBCL diagnosis pathologically with the support of immunohistochemistry. Technically, we could include immunohistochemistry result in AI training if necessary but in our current study we just aimed to distinguish DLBCL from non-DLBCL by only reading the HE tissue slides to show whether we could achieve a high diagnostic accuracy. Indeed, our results showed that our AI models can reach 100% diagnostic accuracy for DLBCL. Again, the reviewer’s suggestion for including immunohistochemistry into our AI model building is outstanding, and in our future study, we will do so to help to sub-classify DLBCL using the AI system.

The reviewer mentioned: *“...it is misleading to test the system against diagnosis done only on HE: this is not the normal clinical practice”*. It is not the normal clinical practice because a pathologist does not diagnose DLBCL solely based on looking at a HE tissue slide. As we pointed out above, AI does not and should not follow what a pathologist does simply because an AI machine does not think and do things in the same way as a pathologist does. The AI machine can detect what a human pathologist can’t. Just because of this, a normal practice by AI in the near future should differ from the *“the normal clinical practice”* by pathologists. The fact is that our AI model can help to diagnose DLBCL only based on HE, which is why we believe that AI pathology has a bright future in helping pathologists in reducing workload and providing correct diagnosis.

The authors write “In clinical practice, a diagnostic accuracy of 100% or greater than 99% is critical and this level of accuracy has not been reported for pathological diagnosis of any types of hematopoietic malignancies.”

This is rather inaccurate and not supported by any evidence. For example I had a very quick look at our double reporting records and, to a very rough analysis, we have an agreement between two or more pathologists on the diagnosis of DLBCL of 98.35, of 98.7% in the

diagnosis of Follicular lymphoma

We apologize for not describing what we really wanted to express clearly. We wanted to say that it will be difficult to provide a diagnosis with 100% correct rate by a pathologist if he/she only looks at a 400x slide. A pathologist often needs to look at many magnifications often including from 200x to 400x original magnification as well as immunohistochemistry and often flow cytometry to provide a final diagnosis. In this revised manuscript, we more clearly described what we really want to express.

Minor issues include:

Line 150: why use only 400x? It could miss the presence of some low grade component in the same slides

As we described above in response to the reviewer, at only 400x, human pathologists may miss ***“the presence of some low grade component in the same slides”*** and other features but AI model will not. This is because the AI model is able to extract thousands or more features from a slide image, whereas a pathologist only focuses on a limited amount of information related to some major morphological changes of diseased cells and their locations. Actually, AI does not know what ***“low grade component”*** is, and AI only picks up all differences between DLBCL and non-DLBCL to be able to identify which image is DLBCL or non-DLBCL. Again, we should not expect or require AI to work in the same ways as pathologists do in their medical practice, because AI does not think and work like a human being. Importantly, AI utilizes its unique way to pick up disease-specific information that we cannot pick up and understand, which is why we need AI in medicine including pathology.

Reviewer #3 (Remarks to the Author):

In their manuscript, Li et al. propose a deep-learning-based platform trained for the diagnosis of diffuse large B cell lymphoma (DLBCL) and non-DLBCL from pathologic images.

With the technique proposed, the authors claim to achieve intra-dataset accuracies between 99.7-100% on data from 3 different hospitals, and inter-dataset accuracies between 80-82%.

We need to make a correction regarding ***“inter-dataset accuracies between 80-82%”*** because the inter-dataset accuracy is actually greater than 90%. During revising our work, we discovered a mismatch of the input data in our examination of ***“inter-dataset accuracies”***, and after correcting this mismatch, the inter-dataset accuracy reached 90.50%, which is among the best in the published studies. We have added this new information to our revised manuscript. By the way, the mismatch was that we did not normalize the images between hospitals into the same shape, which was not appropriate for comparison using an AI system.

The novelty resides in using a combination of 17 trained convolutional neural networks (CNN), instead of comparing performance and choosing the best one as often done. They claim it's the first time a perfect accuracy is reached, which, the way it is stated, is incorrect (see for example, "Deep Convolutional Neural Networks Enable Discrimination of Heterogeneous Digital Pathology Images"). Better setting of the state of the art could be done and the claim phrased more carefully and more precisely; 100% is achieved on intra-dataset test set.

We greatly appreciate the recognition of the novelty of our 17 CNN approach by the reviewer and apologize for the unclear message we described. As for our claim “*it's the first time a perfect accuracy is reached*”, what we actually meant was that it is the first time for us to reach a perfect accuracy in AI diagnosis of DLBCL in an intra-dataset test. In our revised manuscript, we made our point more clearly.

While the results are impressive and the method very interesting, the method lacks details to allow readers be convinced. More specifically, the code should be made available to the reviewers (and later to the readers) so that we could properly assess the work. I can't comment on the well documented sources.

Although we thought we provided sufficient method-related information, during revising our manuscript we tried our best to guess what detailed information the reviewer is asking and added more information in the Method section.

Sorry to say that based on our institutional policy related to the protection of intellectual property (IP), we have no right to disclose the source code prior to publication. All of our AI models have been trained, tested and programmed with Matlab software package and Microsoft Visual Studio. We will certainly make the source code and datasets available to public (readers) after our work is accepted for publication, but if the reviewer were still not satisfied with this solution, we would ask that the reviewer agrees to sign a confidentiality agreement with us for approval of releasing the information by our university.

A few important details needed that would help the readers would be:

**** Transfer learning is used: what were the networks initially trained on? Where they all pre-trained one after the others with their own outputs, or the 17 networks trained as one with a "merged" output?***

The networks were initially trained on ImageNet.

The 17 pre-trained networks were trained as one system with a "merged" output, which is why we emphasized in our manuscript that we combined 17 CNNs.

**** The authors mention variations of pathologic images between hospitals and also presence of artifacts, and they: "We expect that these variations pose an obstacle to establishing highly accurate AI models, which we should pay much attention to when generating our AI models." but don't explain properly how they dealt with it***

We apologize for leaving the reviewer with this concern, and in our revised manuscript we provided more clear strategies for dealing with these variations as described here as well:

1) Building more powerful AI models. In this regard, we built our AI models that are among the best available. As mentioned above, our inter-dataset test, in which the model A established using tissue images of patients from hospital A was used to read the images of patients from hospital C, reached a diagnostic accuracy of 90.50% for DLBCL. This 90.50% inter-dataset accuracy is about 10% drop from our 100% intra-dataset accuracy, contrasting sharply to about 20% drop in accuracy in a beautiful representative study on solid tumors (Campanella, G. *et al.* Clinical-grade computational pathology using weakly supervised deep learning on whole slide images. *Nat Med* 25, 1301-1309, 2019).

2) Standardizing slide preparation procedures and image collection equipment used. The technical variability introduced by slide preparation and image collection equipment could only be partially overcome through improving the building of AI models as we were able to do. A complete elimination of this variability requires different hospitals/institutions/centralized referral centers to use the same slide preparation procedures and the imaging equipment. This strategy has been accepted in the medical research field and we have had experimental evidence to support this strategy. Specifically, we used our AI model established in hospital B to read slide images from another hospital (hospital D) where the tissue slides were prepared using a similar procedures as used in hospital B and the slide images were collected using the same scanner. Our result showed that a 100% diagnostic accuracy was achieved for DLBCL (see new Fig. 3c in our revised manuscript and above in response to the reviewer 1).

3) Building hospital-based AI models. Realistically, we can only ensure the use of the same slide preparation procedures and image collection equipment within a single hospital to avoid the introduction of the technical variability between hospitals into the generation of AI models. This strategy requires an ability to use a smaller dataset (one should not expect to obtain a large number of human samples for a particular disease from any single hospital) to build an AI model with a high diagnostic accuracy, as we did in this study.

**** line 150 "were correctly labeled by pathology experts". I would suggest to be more specific (here or in method): How many experts? What is their level of experience? Did they annotations overlap and if so, was variability assessed?***

We apologize for not providing more details regarding how the diagnosis of each case was determined. Each tissue slide used in our study came to us with a diagnosis and was looked at again and confirmed by at least one pathologist with sufficient experience in hemopathology. We further emphasized this point in our revised manuscript. Controversial slides were eliminated and not used in generating our AI models.

**** "photographed at 400x original magnification". The resulting pixel size should be given to be able to compare with scanned images available on other databases for example.***

We totally agree with the reviewer, and in our revised manuscript we added the following information:

Photos taken from Hospital A:

Objective magnification: 40x

Original image: 2592x1944 pixels

Whole slides information from Hospital B:

Average slide dimensions in pixels: ~200000 x ~400000

Average file size: ~10Gb

Objective magnification: 40x

Micrometer/pixel X: 0.121547

Micrometer/pixel Y: 0.121547

Cropped image for classification at 945x945 pixels

Photos taken from Hospital C:

Objective magnification: 40x

Original image at 2048x1536 pixels

Cropped image at 1075x1075 pixels

**** the authors say the images were split into training/validation/test set. If it is the case and one patient has several images, then, it is possible that a patient has some of his slides in the training set, and some in the test set? The cross-validation accuracy (which is the one that matters) shows the accuracy drops from 99.7-100% to 80-82%, which is a big drop. It either suggests that the slides are very different, or that there is a patient leak when tested on the same hospital's dataset. Please double check and show which case it is. One way to unambiguously prove the sample preparation is the reason for that drop would be to (re)strain slides from a subset of patients using the procedure of the different hospitals. If not, showing the list of patients in each dataset is different is another option.***

The reviewer is correct, and it is possible that a patient has some of his slides in the training set and some in the test set.

The reviewer suggested two possible reasons for “***the accuracy drops from 99.7-100% to 80-82%***” (about 20% drop) in our cross-validation, but now we know that the main reason for the dropping is actually the technical variability introduced by slide preparation procedures and image collection equipment, which were different between hospitals. Before we show our supporting evidence for this reasoning, we would like to make a correction on the 20% drops in our cross-hospital accuracy tests using our AI models. In our originally-submitted manuscript, we had an error in designing our cross-hospital tests. First of all, we inappropriately used the model B established in hospital B to read the images from patients in hospital A to test the generalizable level of our model, which showed about 20% accuracy drop from 99.71% to 79.80%. This test was inappropriate because the slide images in hospital B were collected using a scanner, whereas the slide images in hospital A were collected using a camera. It has been reported that about 3% accuracy drop was observed when two different scanners were used between hospitals in a study of solid tumors (Campanella, G. *et al.* Clinical-grade computational pathology using weakly supervised deep learning on whole slide images. *Nat Med* 25, 1301-1309, 2019). We used totally different types of image collect equipment (scanner vs camera) in the two hospitals, so the percentage of accuracy drop could be even greater. Secondly, although it was appropriate to use the model A established in hospital A to read the images from patients in hospital C because the images from both hospitals were collected in the same way (by photographing), we ignored the fact that the shape of the input images in hospital A were different from those in hospital C, which would negatively affect the diagnostic accuracy, although this should not be considered as a real “mismatch error” because we originally focused on using an AI model within the same hospital where patient samples were obtained and the model was built, which is why we made this “mismatch error” when conducting the cross-hospital test involving hospitals A and C. After we normalized the shape of 179 images in hospital C to those in hospital A and re-did this cross-hospital test, we were able to significantly

increase the diagnostic accuracy from 82.09% to 90.50% (see new Fig. 3c in the revised manuscript and also above in response to the reviewer 1). In our knowledge, we have rarely seen a published work showing this high level of accuracy (90.50%) for diagnosing human cancer in cross-validation using an AI model. Now we should answer the reviewer's question: What caused the accuracy drop from 100% to about 90.50 % when we used our AI model in cross-validation?

Although the 90.50% diagnostic accuracy from our cross-hospital test is among the best, which showed about 10% accuracy drop in contrast to about 20% drop in the solid tumor study (Campanella, G. *et al.* Clinical-grade computational pathology using weakly supervised deep learning on whole slide images. *Nat Med* 25, 1301-1309, 2019), we agree with the reviewer that we should try to identify the factors that prevented us from reaching, in this cross-hospital test, an accuracy close to 100% as we achieved in our intra-hospital accuracy tests within each hospital. We believe that the differences between hospitals in slide preparation procedures and image collection equipment are to blame. This reasoning is based on the published solid tumor work showing that the technical variability introduced by differences in slide preparation and type of scanner caused about 9% drop in diagnostic sensitivity (Campanella, G. *et al.* Clinical-grade computational pathology using weakly supervised deep learning on whole slide images. *Nat Med* 25, 1301-1309, 2019). To provide more supporting evidence for our reasoning, we conducted two new tests. First, we used the model A established in hospital A to read the slide images of new patients from the same hospital with an attempt to eliminate the potential accuracy drop caused by the technical variability introduced by slide preparation procedures and image collection equipment used. Our result showed that the 100% diagnostic accuracy for DLBCL was maintained. Second, we reached out a new hospital (hospital D) that was not involved originally when we submitted our previous manuscript reviewed by the reviewer, because this hospital utilized slide preparation procedures that were similar to the ones used in hospital B. Also, we scanned the pathologic slides from hospital D to produce whole slide images using the same scanner we used to collect whole slide images from hospital B. Thus, we basically eliminated the differences between the two hospitals (hospitals B and D) in slide preparation and image collection equipment. We then used the model B established in hospital B to read the images of patients from hospital D, and 100% diagnostic accuracy for DLBCL was achieved in this inter-hospital test. This new cross-hospital inter-dataset test allowed us to draw two conclusions:

1) We provided the strong evidence showing that diagnostic accuracy by AI can be negatively affected by slide preparation procedure and image collection equipment used, calling for having an international effort to standardize slide preparation procedure and image collection equipment in computational pathology for diagnosing human diseases.

2) We have had an ability to generate powerful AI models that could be soon employed in medical practice to assist pathologists in reducing their intensive workload and helping to provide more accurate diagnosis.

**** Regarding the validation, it looks like a bunch of parameters were tested and optimized. It could be interesting, in a supp table, to show how each parameter tested affected the validation performances and led to the selected optimal conditions.***

We thank the reviewer for this good suggestion. From our work, we learned that AI diagnostic accuracy is mainly affected by slide preparation, image collection equipment, and

image annotation. Thus, we think that a supp table would not be necessary, but we emphasized this information more clearly in our revised manuscript.

**** Authors use "accuracy" only to assess the performance. Other measures are usually more common and could be considered (AUC for example).***

We chose “*accuracy*” because we believe that diagnostic accuracy is the most important and more meaningful criterion for medical diagnosis as it clearly shows how many patients would be diagnosed and mis-diagnosed. AUC (area under the curve) has been widely used by researchers to help to evaluate the overall performance of a probabilistic classifier. However, the models we have built in medical practice must be discrete classifiers (Tom Fawcett, ROC Graphs: Notes and Practical Considerations for Researchers, Pattern Recognition Letters 31(8):1-38 · January 2004). AUC does not accurately eliminate false-negative diagnosis, and in addition, it increases false-positive diagnosis as shown in the representative deep learning study (Campanella, G. *et al.* Clinical-grade computational pathology using weakly supervised deep learning on whole slide images. *Nat Med* 25, 1301-1309, 2019). In contrast, we emphasize the diagnostic accuracy with a goal of eliminating both false-negative and false-positive diagnoses.

**** What is the ground truth /gold standard used to assign the true label to each patient? If it was based on hematopathologists diagnosis, please how differences between pathologists were tackled to assign ground truth.***

We thank the reviewer to ask this important question and we apologize for not making this clear enough in our originally-submitted manuscript. In our view, the “*ground truth /gold standard used to assign the true label to each patient*” includes the following:

- 1) Read the original diagnosis by pathologists correctly.
- 2) Invite at least one experienced pathologist to confirm the original diagnosis by looking at the tissue slide and reviewing the results from other tests including immunohistochemistry, molecular biology, etc.
- 3) Confirm further the original diagnosis by validating the consistency between the pathological diagnosis and patient’s clinical symptoms for the disease.

In this revised manuscript, we made these points more clearly.

**** How are the outputs of all the trained networks combined to generate the final decision score?***

The final score is determined by voting among the 17 CNNs for the result supported by the majority of the class. For example, if 9 out of 17 CNNs classify an image as DLBCL and 8 out of 17 CNNs classify the same image as non-DLBCL, then the image will be finally classified as a DLBCL. There are mathematical formulas in our manuscript to express such a strategy accurately.

**** Which network(s) contribute the most/the best to the output, from a statistical point of view, and from "workbench" tests? It would be good to show the performance of each individual network to solve this task to prove the benefit of combining the 17 of them.***

We do not know which network(s) contribute the most/the best to the output, because our model was treated as a black box, which is how deep learning should work. However, what we do know is that the output of combining the 17 CNNs is much better than any individual network. In fact, we initially used each of the 17 CNNs, respectively, to analyze the pathologic images of

DLBCL in each of the three hospitals, and found that the average diagnostic accuracy in the three hospitals by using one CNN was ranged from 87% to 96%. In our view, the diagnostic accuracy needs to be 100% or greater 99% prior to employing any deep learning model in medical practice. This is why we decided to combine 17 CNNs into one system with our algorithms with a goal of enhancing the performance of deep learning to reach 100% diagnostic accuracy, which we had achieved in our study. We added this information to our revised manuscript.

*** *The different networks used deal with different input sizes. How did the authors deal with it?***

This is a very good question. The 17 pre-trained CNNs have different input image sizes, such as 224x224, 227x227, 299x299, and 331x331. Those requirements were met by calling resize functions respectively according different networks in our platform.

*** *The links to the image and code are not yet working. It is not possible to assess the feasibility of the technique without having a view on the code. Furthermore, as often requested by most journals, code must be properly commented and written in a way that will allow users to reproduce the results.***

Although we thought we provided sufficient method-related information, in our revised manuscript we tried our best to guess what detailed information the reviewer is asking for and added more information in the Method section.

Again, sorry to say that based on our institutional policy related to the protection of intellectual property (IP), we have no right to disclose the code prior to publication. We will certainly make the source code available to public (readers) after our work is accepted for publication. However, if the reviewer insists in reviewing the code at this early stage, as we suggested above, we would ask that the reviewer agrees to sign a confidentiality agreement with us for approval of releasing the information by our university.

As a conclusion, I think the main interesting point of this paper is the fact that they combine 17 different neural networks to achieve the classification. However, the lack of precision in the method, the lack of intermediate results and lack of analysis of the obtained results cast a significant shadow on the current manuscript. I hope the authors will solve the concerns and questions addressed because if they show there is no patient leak and the 17-network combination performs significantly better than each individual ones, then it would be a very interesting paper.

Again, we thank the reviewer for acknowledging the novelty of our work using combined 17 CNNs in our study. In this revised manuscript, we provided more information and conducted new tests to address the reviewer's concerns on "***the lack of precision in the method, the lack of intermediate results and lack of analysis of the obtained results***". We hope that now the reviewer would feel satisfied with our revisions and we thank the reviewer again for raising so many good questions to help to strengthen our work.

Reviewers' Comments:

Reviewer #1:

Remarks to the Author:

No further comments.

Reviewer #2:

Remarks to the Author:

I am happy with the authors reply.

Reviewer #3:

Remarks to the Author:

The paper has been considerably improved with the last modifications. There are however a few answers that are incomplete, or would require further clarification:

>> "The reviewer is correct, and it is possible that a patient has some of his slides in the training set and some

>> in the test set."

This information should be clearly stated in the text. Also, in addition to the number of slides for each hospital, the number of patients should be given so the readers can have an idea of how many slides per patient are used.

Ideally, to assess whether the 100% performance is somehow linked to recognition of intra-patient features, tests where slides from patients are separated and do not appear in similar sets should have been done.

Khosravi et al. for example (10.1016/j.ebiom.2017.12.026, who, BTW, also obtained 100% with AI on histopathology images classification and should be discussed somewhere) have shown to some extent that perfect AUC can be obtained on intra-slide classifications. One can expect that if several slides from a same patient are in different sets, the same thing can happen.

>>>> "photographed at 400x original magnification". The resulting pixel size should be given to be able to compare

>>>> with scanned images available on other databases for example.

>> We totally agree with the reviewer, and in our revised manuscript we added the following information:

>> Photos taken from Hospital A:

>> Objective magnification: 40x

>> Original image: 2592x1944 pixels

>> Whole slides information from Hospital B:

>> Average slide dimensions in pixels: ~200000 x ~400000

>> Average file size: ~10Gb

>> Objective magnification: 40x

>> Micrometer/pixel X: 0.121547

>> Micrometer/pixel Y: 0.121547
>> Cropped image for classification at 945x945 pixels
>> Photos taken from Hospital C:
>> Objective magnification: 40x
>> Original image at 2048x1536 pixels
>> Cropped image at 1075x1075 pixels

The information added regarding the images is not complete. To have a possible comparison, the pixel size must be given for all hospital. Here, it's only given for hospital B.

>>>> Which network(s) contribute the most/the best to the output, from a statistical point of view, and from
>>>> "workbench" tests? It would be good to show the performance of each individual network to solve this task to
>>>> prove the benefit of combining the 17 of them.
>> We do not know which network(s) contribute the most/the best to the output, because our model was treated as
>> a black box, which is how deep learning should work. However, what we do know is that the output of combining the
>> 17 CNNs is much better than any individual network. In fact, we initially used each of the 17 CNNs, respectively,
>> to analyze the pathologic images of DLBCL in each of the three hospitals, and found that the average diagnostic
>> accuracy in the three hospitals by using one CNN was ranged from 87% to 96%. In our view, the diagnostic accuracy
>> needs to be 100% or greater 99% prior to employing any deep learning model in medical practice. This is why we
>> decided to combine 17 CNNs into one system with our algorithms with a goal of enhancing the performance of deep
>> learning to reach 100% diagnostic accuracy, which we had achieved in our study. We added this information to our
>> revised manuscript.
>> "model was treated as a black box, which is how deep learning should work"

(This is not true! It is how deep learning often does work but it is not how it should work. Many groups are actually working on trying to understand how they work and make decisions!)

>> "In fact, we initially used each of the 17 CNNs, respectively, to analyze the pathologic images of DLBCL in each of
>> the three hospitals, and found that the average diagnostic accuracy in the three hospitals by using one CNN was
>> ranged from 87% to 96%."

I believe this could be very interesting to readers. If this was indeed done, then why not include the detailed data in the manuscript? This would be a very interesting comparison and piece of information! I see just the sentence added, but no detailed table with what is the performance of each of these 17 CNNs, not only the range. (please add reference to the table in the main text if the table is there but I missed it)

>>>> How are the outputs of all the trained networks combined to generate the final decision score?

>> The final score is determined by voting among the 17 CNNs for the result supported by the majority of the class. For
>> example, if 9 out of 17 CNNs classify an image as DLBCL and 8 out of 17 CNNs classify the same image as non-DLBCL,
>> then the image will be finally classified as a DLBCL. There are mathematical formulas in our manuscript to express
>> such a strategy accurately.

This answer seems to contradict the one above where the authors said "We do not know which network(s) contribute". Here they say: "if 9 out of 17 CNNs classify an image as DLBCL and 8 out of 17 CNNs classify the same image as non-DLBCL, then the image will be finally classified as a DLBCL.", meaning you do have access to the performance of the 17 network. And since the decision is a simple count, it should be possible to analyze in details the contributions, which ones are significant or not, etc...

>> Other comments:

In the abstract, the authors added:

"The 100% accuracy was maintained after eliminating the technical variability between hospitals"

This seems wrong and misleading since it could be understood as the inter-hospital AUC is 100%, but it's not, it's ~90%. It should be made very clear in the abstract that 100% in the intra-hospital performance and 90% the inter-hospital one.

Point-by-point response to reviewers

We thank the reviewers for their time and patience in reviewing our manuscript. Their comments and suggestions are extremely valuable and have helped to improve our manuscript greatly. Our point-by-point response to the reviewers is shown below:

REVIEWER COMMENTS

Reviewer #1 (Remarks to the Author):

No further comments.

Response: We thank the reviewer for supporting our work.

Reviewer #2 (Remarks to the Author):

I am happy with the authors reply.

Response: We thank the reviewer for supporting our work.

Reviewer #3 (Remarks to the Author):

The paper has been considerably improved with the last modifications. There are however a few answers that are incomplete, or would require further clarification:

“The reviewer is correct, and it is possible that a patient has some of his slides in the training set and some in the test set.”

This information should be clearly stated in the text. Also, in addition to the number of slides for each hospital, the number of patients should be given so the readers can have an idea of how many slides per patient are used.

Response: We thank the reviewer for this important comment and had described this information more clearly in our revised manuscript. Also, we had more clearly described the number of patients in each hospital in our revised manuscript.

Ideally, to assess whether the 100% performance is somehow linked to recognition of intra-patient features, tests where slides from patients are separated and do not appear in similar sets should have been done.

Response: As described in our manuscript, we used 0.8-0.1-0.1 format (training 0.8, validation 0.1, testing 0.1). In other words, we purposely separated 10% of all tissue images from patients in each of the three hospitals for testing each of our AI models. For example, in hospital A, we used 90% of photographed tissue images for training and validation, and 10%, for testing. To emphasize, those 10% images were not involved in training and validation of the AI model. The same was done for hospital C. Therefore, we are confident that the 100% performance of our AI models truly reflects the ability of our models in identifying the pathological features of DLBCL.

Khosravi et al. for example (10.1016/j.ebiom.2017.12.026, who, BTW, also obtained 100% with AI on histopathology images classification and should be discussed somewhere) have shown to some extent that perfect AUC can be obtained on intra-slide classifications. One can expect that if several slides from a same patient are in different sets, the same thing can happen.

Response: Compared to the whole slide images in hospital B where “several slides from a same patient are in different sets”, in hospitals A and C, each patient had only one photographed tissue image, so the situation “several slides from a same patient are in different sets” did not exist in those two hospitals. Thus, our 100% performance of our AI

models truly reflects the ability of our models in identifying the pathological features of DLBCL.

We thank the reviewer for pointing out the beautiful paper by Khosravi, although this paper focused on a type of solid tumor. To emphasize the potential of AI technology in computational pathology, we have cited this paper in our revised manuscript. We could not further discuss the results in the paper by Khosravi, because data and code were not provided in the paper to allow a test-run on the generalization of the deep learning model used in this published study.

"photographed at 400x original magnification". The resulting pixel size should be given to be able to compare with scanned images available on other databases for example.

We totally agree with the reviewer, and in our revised manuscript we added the following information:

Photos taken from Hospital A:

Objective magnification: 40x

Original image: 2592x1944 pixels

Whole slides information from Hospital B:

Average slide dimensions in pixels: ~200000 x ~400000

Average file size: ~10Gb

Objective magnification: 40x

Micrometer/pixel X: 0.121547

Micrometer/pixel Y: 0.121547

Cropped image for classification at 945x945 pixels

Photos taken from Hospital C:

Objective magnification: 40x

Original image at 2048x1536 pixels

Cropped image at 1075x1075 pixels

The information added regarding the images is not complete. To have a possible comparison, the pixel size must be given for all hospital. Here, it's only given for hospital B.

Response: Following the reviewer's suggestion, we provided the requested information in our revised manuscript as follows:

Photos taken from Hospital A:

Objective magnification: 40x

Original image: 2592x1944 pixels

Original image file size: 14.4Mb

Pixel Size: 2.2 μ m x 2.2 μ m

Whole slides information from Hospital B:

Average slide dimensions in pixels: ~200000 x ~400000

Average file size: ~10Gb

Objective magnification: 40x

Pixel size: 0.121547 μ m x 0.121547 μ m

Cropped image for classification at 945x945 pixels

Photos taken from Hospital C:

Objective magnification: 40x

Original image: 2048x1536 pixels

Original image file size: 5-8Mb

Pixel Size: 3.45 μ m x 3.45 μ m

Cropped image at 1075x1075 pixels

Which network(s) contribute the most/the best to the output, from a statistical point of view, and from "workbench" tests? It would be good to show the performance of each individual network to solve this task to prove the benefit of combining the 17 of them.

We do not know which network(s) contribute the most/the best to the output, because our model was treated as a black box, which is how deep learning should work. However, what we do know is that the output of combining the 17 CNNs is much better than any individual network. In fact, we initially used each of the 17 CNNs, respectively, to analyze the pathologic images of DLBCL in each of the three hospitals, and found that the average diagnostic accuracy in the three hospitals by using one CNN was ranged from 87% to 96%. In our view, the diagnostic accuracy needs to be 100% or greater 99% prior to employing any deep learning model in medical practice. This is why we decided to combine 17 CNNs into one system with our algorithms with a goal of enhancing the performance of deep learning to reach 100% diagnostic accuracy, which we had achieved in our study. We added this information to our revised manuscript. "model was treated as a black box, which is how deep learning should work"

(This is not true! It is how deep learning often does work but it is not how it should work. Many groups are actually working on trying to understand how they work and make decisions!)

Response: we apologize for not making our point clear, and we agree with the reviewer that the model was treated as a black box, which is how deep learning often works.

"In fact, we initially used each of the 17 CNNs, respectively, to analyze the pathologic images of DLBCL in each of the three hospitals, and found that the average diagnostic accuracy in the three hospitals by using one CNN was ranged from 87% to 96%."

I believe this could be very interesting to readers. If this was indeed done, then why not include the detailed data in the manuscript? This would be a very interesting comparison and piece of information! I see just the sentence added, but no detailed table with what is the performance of each of these 17 CNNs, not only the range. (please add reference to the table in the main text if the table is there but I missed it)

Response: Originally, we did not think it would be necessary to include a table showing "what is the performance of each of these 17 CNNs" in order for us to focus on the performance of our combined 17 CNNs. Following the reviewer's request, we added a new table as Table 1 in the main text of our revised manuscript. For the reviewer's convenience, we attached the table here:

CNNs	Diagnostic Accuracy %		
	Model A	Model B	Model C
AlexNet	92.08	93.57	95.12
GoogleNet	95.05	90.68	95.12
Vgg16	95.05	94.53	99.50
ResNet18	92.08	88.42	95.12
SqueezeNet	92.08	89.39	92.68
MobileNetv2	90.10	88.42	92.68
Inceptionv3	90.10	93.89	87.80
DenseNet201	90.10	84.57	95.12
Xception	98.02	91.32	85.37
Vgg19	87.13	93.25	92.68
Places365GoogleNet	96.04	92.93	95.12

InceptionResNetv2	94.06	96.14	96.02
ResNet50	86.14	90.68	87.80
ResNet101	89.11	91.96	97.56
NASNetMobile	95.05	85.21	90.24
NASNetLarge	95.05	91.96	92.50
ShuffleNet	87.13	88.42	85.37
GOTDP-MP-CNNs (with combined 17 CNNs)	100.00	99.71	100.00

How are the outputs of all the trained networks combined to generate the final decision score?

The final score is determined by voting among the 17 CNNs for the result supported by the majority of the class. For example, if 9 out of 17 CNNs classify an image as DLBCL and 8 out of 17 CNNs classify the same image as non-DLBCL, then the image will be finally classified as a DLBCL. There are mathematical formulas in our manuscript to express such a strategy accurately.

This answer seems to contradict the one above where the authors said “We do not know which network(s) contribute”. Here they say: “if 9 out of 17 CNNs classify an image as DLBCL and 8 out of 17 CNNs classify the same image as non-DLBCL, then the image will be finally classified as a DLBCL.”, meaning you do have access to the performance of the 17 network. And since the decision is a simple count, it should be possible to analyze in details the contributions, which ones are significant or not, etc...

Response: To add more explanations to our response to the previous comment by the reviewer in this rebuttal letter, we said “We do not know which network(s) contribute the most/the best to the output” because we were not interested in any individual CNN’s performance or we did not strongly depend on any individual CNN’s performance. Therefore, we did not show any individual CNN’s performance in our previous revised manuscript. As mentioned above, now we have added the table showing the individual performance of 17 CNNs in the main text of our revised manuscript.

Other comments:

In the abstract, the authors added:

“The 100% accuracy was maintained after eliminating the technical variability between hospitals”

This seems wrong and misleading since it could be understood as the inter-hospital AUC is 100%, but it’s not, it’s ~90%. It should be made very clear in the abstract that 100% in the intra-hospital performance and 90% the inter-hospital one.

Response: We apologize for leaving the reviewer a wrong impression about the performance of our model when doing a cross-hospital test. We simply attempted to emphasize and send an important message that the reduced performance (by about 10%) is caused by the technical variability introduced by slide preparation methods and imaging equipment caused, and if this technical variability is eliminated, the performance is back to 100%. Keeping this in mind, we have revised the abstract in our revised manuscript by saying the following:

“Although the technical variability introduced by slide preparation and image collection reduced AI model performance in cross-hospital tests, the 100% diagnostic accuracy was maintained after eliminating this variability.”

Reviewers' Comments:

Reviewer #3:

Remarks to the Author:

Thanks for the additions and clarifications. I have no other comment.

This is a nice work. Once published, please make the code available for the community with clear guidelines on its usage.